# DUAL-ENCODERS FOR EXTREME MULTI-LABEL CLASSIFICATION

**Nilesh Gupta**[†◇*] **Devvrit Khatri**[†◇] **Ankit Singh Rawat**[‡] **Srinadh Bhojanapalli**[‡]
**Prateek Jain**[‡] **Inderjit Dhillon**[†◇]
[†]The University of Texas at Austin [◇]Google [‡]Google Research

## ABSTRACT

Dual-encoder (DE) models are widely used in retrieval tasks, most commonly studied on open QA benchmarks that are often characterized by multi-class and limited training data. In contrast, their performance in multi-label and data-rich retrieval settings like extreme multi-label classification (XMC), remains under-explored. Current empirical evidence indicates that DE models fall significantly short on XMC benchmarks, where SOTA methods (Dahiya et al., 2023a;b) linearly scale the number of learnable parameters with the total number of classes (documents in the corpus) by employing per-class classification head. To this end, we first study and highlight that existing multi-label contrastive training losses are not appropriate for training DE models on XMC tasks. We propose decoupled softmax loss – a simple modification to the InfoNCE loss – that overcomes the limitations of existing contrastive losses. We further extend our loss design to a soft top-k operator-based loss which is tailored to optimize top-k prediction performance. When trained with our proposed loss functions, standard DE models alone can match or outperform SOTA methods by up to 2% at Precision@1 even on the largest XMC datasets while being 20× smaller in terms of the number of trainable parameters. This leads to more parameter-efficient and universally applicable solutions for retrieval tasks. Our code and models are publicly available here.

## 1 INTRODUCTION

Dual-encoder (DE) models have been highly successful in dense retrieval tasks for open-domain question answering (openQA) systems (Lee et al., 2019; Karpukhin et al., 2020; Qu et al., 2021), where they efficiently retrieve relevant passages or documents from a large corpus given a user's query. These models map both queries and documents into a shared embedding space, enabling efficient retrieval using *fast similarity search methods* (Guo et al., 2020; Johnson et al., 2021). Notably, openQA benchmarks are often characterized by multi-class (most queries are tagged with a single positive document) and limited (each document in the corpus has zero or very few tagged queries i.e. *zero/few-shot scenario*) training data. On these benchmarks, models are required to *generalize* and retrieve relevant documents even though they might not have appeared in the training set.

Another important retrieval scenario that frequently arises in real-world applications such as search engines (Mitra & Craswell, 2018) and recommendation systems (Covington et al., 2016; Zhang et al., 2019) is where we want to perform retrieval for multiple documents/items from a large collection based on a significant number of seen examples per document/item, i.e., a *many-shot scenario*. Such tasks are often formulated as extreme multi-label classification (XMC) tasks, where each document/item to be retrieved is considered as a separate label (Bhatia et al., 2016). XMC also appears in other emerging applications such as retrieval augmented generation (RAG) (Lewis et al., 2020). In particular, when multiple documents can potentially contain the desired answer, the retrieval task in RAG closely aligns with an XMC problem. Typically, XMC algorithms need to both "memorize" and "generalize". That is, for each label they need to memorize the type of queries that are relevant; e.g., for a product to product recommendation scenario, the algorithm should memorize which products can lead to click on a particular product using the provided product-product co-click data. At the same time, the algorithm should generalize on unseen queries.

---

[*]Correspondence: `nilesh@cs.utexas.edu`

It is a prevailing belief in the XMC community that due to the semantic gap and knowledge-intensive nature of XMC benchmarks, DE by themselves are not sufficient to attain good performance (Dahiya et al., 2023b). As a measure to overcome the semantic gap and enable memorization, state-of-the-art (SOTA) XMC methods augment DE with either per-label classifiers (Dahiya et al., 2023a; Mittal et al., 2021a) or auxiliary parameters (Dahiya et al., 2023b). We explore this belief by performing a simple experiment on a synthetic dataset where a random query text is associated with a random document text, and the task is to memorize these random correlations during retrieval. We find that DE models are able to perform this task with perfect accuracy at least on up to 1M scale datasets (cf. Section C.1), disputing the previously held belief in the literature.

In light of the aforementioned synthetic experiment, this work aims to answer the following question: "Are pure DE models sufficient for XMC tasks?" A Pure DE model is desirable because: 1) unlike XMC methods, DE methods are parameter efficient i.e. the model size (consequently the number of trainable parameters) does not scales *linearly* with the number of labels (see Figure 1). 2) XMC methods require re-training or model updates on encountering new labels. In contrast, DE methods can generalize to new labels based on their features. Interestingly, our work shows that pure DE models can indeed match or even outperform SOTA XMC methods by up to 2% even on the largest public XMC benchmarks while being 20× smaller in model size. The key to the improved performance is the right loss formulation for the underlying task and the use of extensive negatives to give consistent and unbiased loss feedback. We refer to DE models trained with the proposed approach as **DEXML** (Dual-Encoders for eXtreme Multi-Label classification).

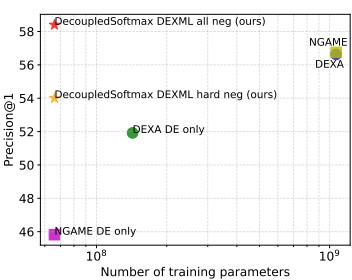

Figure 1: Number of trainable parameters used by different models and their Precision@1 performance on LF-AmazonTitles-1.3M dataset (Bhatia et al., 2016)

We begin by analyzing existing multi-label and contrastive training losses like One-versus-All Binary Cross-Entropy (OvA-BCE) and InfoNCE, highlighting how they may be inadequate for training dual-encoder models in XMC. Specifically, we note that OvA-BCE does not train effectively and InfoNCE disincentivizes confident predictions. To overcome these limitations, we propose a simple modification to the InfoNCE loss as the DecoupledSoftmax loss which removes the undesirable correlation between positive labels during training. We further extend our loss design to SoftTop-k loss which is tailored to optimize prediction accuracy within a specific prediction budget size (k). This can be particularly relevant to XMC applications where the goal is to make a fixed number of highly accurate predictions from a vast set of possible labels. Moreover, we note that XMC problems involve a long tail of labels, which at the query level manifests as a large number of highly relevant negatives. To this end, in order to establish the full potential of DE methods, we first study our proposed loss using *all* the negatives in the loss term. Naturally, including all negatives is challenging for million-label scale datasets. To this end, we provide a memory-efficient distributed implementation that can use multiple GPUs and a modified gradient cache (Gao et al., 2021) approach to scale DE training to largest public XMC benchmarks even with a modest GPU setup (see Table 1, 9). We further study the performances and implications of approximations of the proposed loss functions using standard hard negative mining approaches. In summary, the contributions of this work include:

- We demonstrate that dual-encoder methods alone can achieve SOTA performance on XMC tasks leading to a more parameter-efficient and generalizable approach for XMC. This also unifies the approaches for XMC and retrieval problems, paving the way for simplified and universally applicable solutions.
- We analyze and show existing multi-label and contrastive training losses are not appropriate for training dual-encoder models in XMC settings. Specifically, we note that OvA-BCE does not train effectively and InfoNCE disincentivizes confident predictions.
- We propose a simple modification to the InfoNCE loss, namely DecoupledSoftmax loss, which overcomes the limitations. We further extend our loss design to SoftTop-k loss which is tailored to optimize prediction accuracy within a specific prediction budget size.
- Lastly, we explore applications of our proposed loss functions on an RAG task in Section C.9 which shows promise for our proposed approach over standard encoder training in RAG.

Before providing a detailed account of our contributions in Section 4 and 5, we begin by discussing relevant literature in Section 2 and introducing necessary background in Section 3.

## 2 RELATED WORK

**Extreme classification methods.** There is a large body of literature on utilizing sparse features or dense pre-trained dense features to develop XMC methods (Babbar & Schölkopf, 2017; Prabhu et al., 2018; Prabhu & Varma, 2014; Joulin et al., 2017; Yen et al., 2016; 2017; Gupta et al., 2021; Yu et al., 2022). Lately, task-specific dense features, e.g., based on BERT (Devlin et al., 2019), have been also utilized to design solutions for XMC problems (see, e.g., Zhang et al., 2018; You et al., 2019; Gupta et al., 2019; Guo et al., 2019; Medini et al., 2019; Chang et al., 2020; Jiang et al., 2021; Ye et al., 2020; Zhang et al., 2021; Gupta et al., 2022, and references therein). However, these approaches treat labels as featureless ids and employ classification networks that assign a unique classification vector to each label. Given their increasingly popularity in information retrieval literature (Lee et al., 2019; Karpukhin et al., 2020; Luan et al., 2021; Qu et al., 2021; Xiong et al., 2021), DE methods that employ two encoders to embed query and items into a common space have been increasingly explored for XMC settings as well. By leveraging the label features such as label text (Mittal et al., 2021a; Dahiya et al., 2021), images (Mittal et al., 2022), and label (correlation) graph (Mittal et al., 2021b; Saini et al., 2021), such methods have already shown improved performance for the settings involving tail labels. That said, the current SOTA methods such as NGAME (Dahiya et al., 2023a), do not solely rely on a DE model. For instance, NGAME's two-stage approach involves first training query representations via a DE approach, and then utilizing a classification network in the second stage. DEXA (Dahiya et al., 2023b), is a recently proposed approach which builds on NGAME to improve the encoder embeddings by augmenting them with auxiliary parameters.

**Loss functions for multi-label classification problems.** One of the prevalent approaches to solve multi-label classification problem involves reducing it to multiple (independent) binary (Dembczyński et al., 2010) or multiclass (Joulin et al., 2017; Jernite et al., 2017) classification problems. Subsequently, one can rely on a wide range of loss functions designed for the resulting problems, aimed at minimizing the top-$k$ classification error (Lapin et al., 2016; Berrada et al., 2018; Yang & Koyejo, 2020; Petersen et al., 2022). Since different queries have varying number of relevant documents, Lapin et al. (2016); Yang & Koyejo (2020) identified desirable loss functions that are simultaneously calibrated (Bartlett et al., 2006; Zhang, 2004) for multiple values of $k$. Another popular approach to solve a multi-label classification problem relies on (contextual) *learning to rank* framework (Liu et al., 2009) where (given a query) the main objective is to rank relevant documents above irrelevant ones. Various surrogate ranking loss functions, aimed at optimizing the specific ranking metrics such as Precision@k and NDCG@k, have been considered in the literature (see e.g., Usunier et al., 2009; Kar et al., 2015; Li et al., 2017; Jagerman et al., 2022; Su et al., 2022, and references therein).

**Negative sampling.** Modern retrieval systems increasingly deal with a large number of items, leading to increased training complexity. *Negative sampling* (Bengio & Senecal, 2008; Mikolov et al., 2013a; Mnih & Teh, 2012; Mnih & Kavukcuoglu, 2013) helps mitigate this via employing a loss function restricted to the given relevant documents and a subset of negatives or irrelevant documents. Selection of informative negatives is critical for training high-quality models. For classification networks, Bengio & Senecal (2008) studied negative sampling for softmax cross entropy loss by relying on importance sampling. Recently, many works have extended this line of work, e.g., by leveraging quadratic kernels (Blanc & Rendle, 2018) and random Fourier features (Rawat et al., 2019). As for the generic loss function and model architecture, it is common to employ sparse retrieval mechanism, e.g., BM25 (Robertson et al., 1995; 2004), to obtain negative documents (Luan et al., 2021; Karpukhin et al., 2020). Another strategy is to leverage *in-batch negatives* (Henderson et al., 2017; Karpukhin et al., 2020): for a given query in a mini-batch, items appearing as positive labels for other queries in the mini-batch are treated as negatives. However, in-batch negatives often constitute "easy" negatives (with very small contribution to the overall gradient) and do not aid the training much (see, e.g., Karpukhin et al., 2020; Luan et al., 2021). Same holds for the negatives sampled according to uniform or unigram distribution (Mikolov et al., 2013b). Reddi et al. (2019) proposed sampling a large uniform subset of items and weighting those according to their hardness for each query. Taking this approach to an extreme, for each query, ANCE (Xiong et al., 2021) selects the hardest negatives from the entire corpus according to the query-document similarity based on a document index.

## 3 BACKGROUND: MULTI-LABEL CLASSIFICATION

As discussed in Section 1, the (many-shot) retrieval problem explored in this work is essentially a multi-label classification problem. Assuming that $\mathcal{Q}$ and $\mathcal{D}$ denote the query and document (or

label) spaces, respectively, the underlying *query-document relevance* is defined by the distribution P where $P(d|q)$ captures the true relevance for the query-document pair $(q, d) \in \mathcal{Q} \times \mathcal{D}$. In this paper, we assume that the document space is finite with a total $|\mathcal{D}| = L$ documents. Let the training data consists of $N$ examples $\mathcal{S} := \{(q_i, \mathbf{y}_i)\}_{i \in N} \subseteq \mathcal{Q} \times \{0,1\}^L$, where, for $j \in [L]$, $y_{i,j} = 1$ *iff* $j$th document in $\mathcal{D}$ is relevant for query $q_i$. We also denote the set of positive labels for $q_i$ as $P_i = \{j : y_{i,j} = 1\}$

**Dual-encoder models and classification networks.** Note that learning a retrieval model is equivalent to learning a *scoring function* (or simply *scorer*) $s_{\boldsymbol{\theta}} : \mathcal{Q} \times \mathcal{D} \to \mathbb{R}$, which is parameterized by model parameters $\boldsymbol{\theta}$. A DE model consists of a *query encoder* $f_{\boldsymbol{\phi}} : \mathcal{Q} \to \mathbb{R}^d$ and a *document encoder* $h_{\boldsymbol{\psi}} : \mathcal{D} \to \mathbb{R}^d$ that map query and document features to $d$-dimensional embeddings, respectively. Accordingly, the corresponding scorer for the DE model is defined as: $s_{\boldsymbol{\theta}}(q, d) = \langle x_q, z_d \rangle = \langle f_{\boldsymbol{\phi}}(q), h_{\boldsymbol{\psi}}(d) \rangle$, where $\boldsymbol{\theta} = [\boldsymbol{\phi}; \boldsymbol{\psi}]$ and $x_q, z_d \in \mathbb{R}^d$. Note that it is common to share encoders for query and document, i.e., $\boldsymbol{\phi} = \boldsymbol{\psi}$. We also focus on this shared parameter setting in our exploration.

Different from DE models, classification networks ignore document features. Such networks consist of a query encoder $g_{\boldsymbol{\phi}} : \mathcal{Q} \to \mathbb{R}^d$ and a classification matrix $W \in \mathbb{R}^{L \times d}$, with $i$th row as the classification weight vector for the $i$th document. The scorer for the classification network then becomes $s_{\boldsymbol{\theta}}(q, d) = \langle g_{\boldsymbol{\phi}}(q), \mathbf{w}_d \rangle$, where $\mathbf{w}_d$ denotes the classification vector associated with document $d$ and $\boldsymbol{\theta} = [\boldsymbol{\phi}; W]$. For ease of exposition, we do not always highlight the explicit dependence on trainable parameters $\boldsymbol{\theta}, \boldsymbol{\phi}, \boldsymbol{\psi}$ while discussing DE models and classification networks.

**Loss functions.** Given the training set $\mathcal{S}$, one learns a scorer by minimizing the following objective:

$$\mathcal{L}(s; \mathcal{S}) = \frac{1}{N} \sum_{i \in [N]} \ell(q_i, \mathbf{y}_i; s), \tag{1}$$

where $\ell$ is a *surrogate loss-function* for some specific metrics of interest (e.g., Precision@k and Recall@k). Informally, $\ell(q, \mathbf{y}; s)$ serves as a proxy of the quality of scorer $s$ for query $q$, as measured by that metric with $\mathbf{y}$ as ground truth labels. One of the popular loss functions for multi-label classification problem is obtained by employing one-vs-all (OvA) reduction to convert the problem into $L$ binary classification tasks (Dembczyński et al., 2010). Subsequently, we can invoke a binary classification loss such as sigmoid *binary cross-entropy* (BCE) for $L$ tasks, which leads to OvA-BCE:

$$\ell(q, \mathbf{y}, s) = -\sum_{j \in [L]} \left( y_j \cdot \log \frac{e^{s(q, d_j)}}{1 + e^{s(q, d_j)}} + (1 - y_j) \cdot \log \frac{1}{1 + e^{s(q, d_j)}} \right). \tag{2}$$

Alternatively, one can employ a multi-label to multi-class reduction (Menon et al., 2019) and invoke softmax cross-entropy (SoftmaxCE) loss for each positive label:

$$\ell(q, \mathbf{y}, s) = -\sum_{j \in [L]} y_j \cdot \log \left( e^{s(q, d_j)} / \sum_{l \in [L]} e^{s(q, d_l)} \right). \tag{3}$$

Our empirical findings in Section 5.5 reveal that DE models fail to train with BCE loss. We hypothesize that this might be due to the stringent demand of the BCE loss, which requires all negative pairs to exhibit high absolute negative similarity and all positive pairs to have a high absolute positive similarity – a feat that can be challenging for DE representations. In the section below we analyze SoftmaxCE loss (an extreme case of InfoNCE) in more detail and highlight its shortcomings.

## 4 IMPROVED TRAINING OF DUAL-ENCODER MODELS

In this section, we first discuss the loss function typically used for training dual-encoders, and its shortcomings for multi-label problems. We then suggest simple fixes to address the problems, argue why these changes are necessary, and discuss practical implications of our modifications. Finally, we discuss our proposed soft top-k loss formulation and discuss its implementation.

### 4.1 LIMITATIONS OF STANDARD CONTRASTIVE LOSS FUNCTIONS FOR EXTREME MULTI-LABEL PROBLEMS

Existing dual-encoder methods generally rely on contrastive losses for training. For the rest of the paper, we will anchor our technique, experiments, and discussion on the popular InfoNCE contrastive learning loss (Oord et al., 2018), but the observations and the approach apply to other contrastive learning loss functions considered in Xiong et al. (2021); Dahiya et al. (2021; 2023a). In a multi-label

setting, given a batch of queries $\mathcal{B}$ and their positive labels, InfoNCE loss takes the form:

$$\mathcal{L}(s;\mathcal{B}) = \sum_{q_i \in \mathcal{B}} \ell(q_i, \mathbf{y}_i; s) = -\sum_{q_i \in \mathcal{B}} \sum_{j \in [L]} y_{ij} \cdot \log \frac{e^{s(q_i, d_j)}}{e^{s(q_i, d_j)} + \sum_{d^- \in \mathcal{D}^- \setminus d_j} e^{s(q_i, d^-)}}, \quad (4)$$

where $s(q_i, d_j)$ is the similarity score for query $q_i$ and document $d_j$. $\mathcal{D}^-$ denotes the set of (hard) negatives created for the batch $\mathcal{B}$. That is, for in-batch negatives, $\mathcal{D}^-$ represents union of all positive labels for all the queries $q_i \in \mathcal{B}$. In the extreme case when we consider all labels in $\mathcal{D}^-$, (4) becomes the same as the SoftmaxCE loss in (3). A quick analysis of the gradients of this loss function reveals that:

$$\frac{\partial \ell_i}{\partial z_{d_l}} = \begin{cases} (1 - |P_i|\sigma_{il})x_{q_i} & \text{if } l \in P_i \\ -|P_i|\sigma_{il}x_{q_i} & \text{otherwise} \end{cases} ; \sigma_{ij} = \frac{e^{s(q_i, d_j)}}{\sum_k e^{s(q_i, d_k)}}, \text{where } s(q, d) = x_q^T z_d$$

This implies that the loss function is optimized when all positive labels score $\sigma_{il} = \frac{1}{|P_i|}$, while all negative labels achieve $\sigma_{il} = 0$. While this approach appears to be appropriate for elevating the ranking of positive labels above negatives, it raises concerns when applied to imbalanced real-world XMC datasets. In these datasets, not all positive labels possess equal ease of prediction from the available data. To illustrate this issue, consider a scenario where one positive label, denoted as $l$, is straightforward to predict for a given query $q_i$, while the remaining positive labels are characterized by ambiguity. Consequently, the model assigns a high score to the unambiguous label $l$. However, this leads to an increase in the softmax score for label $l$, resulting in $\sigma_{il}$ exceeding $\frac{1}{|P_i|}$. Consequently, the loss function begins penalizing this particular positive label. Given the inherent label imbalance in XMC datasets, the uniform scoring of all positive labels, and the subsequent penalization of confidently predicted positives, do not align with the ideal modeling strategy.

**Decoupled multi-label InfoNCE formulation.** Based on the above observations, we re-formulate the multi-label InfoNCE loss as:

$$(\text{DecoupledSoftmax}) \quad \ell(q_i, \mathbf{y}_i; s) = -\sum_{j \in [L]} y_{ij} \cdot \log \frac{e^{s(q_i, d_j)}}{e^{s(q_i, d_j)} + \sum_{l \in [L]} (1 - y_{il}) \cdot e^{s(q_i, d_l)}} \quad (5)$$

where $s(q, d)$ is model assigned score to the query-document pair. Note that we *exclude* other positives from the softmax's denominator when computing the loss for a given positive, and add all the negatives in the denominator to eliminate the bias/variance in the estimation of that term. Analyzing the gradients with the proposed change gives:

$$\implies \frac{\partial \ell_i}{\partial z_{d_l}} = \begin{cases} (1 - \sigma_{il})x_{q_i} & \text{if } l \in P_i \\ -\sum_{j \in P_i} \frac{e^{s(q_i, d_l)}}{e^{s(q_i, d_j)} + \sum_{k \notin P_i} e^{s(q_i, d_k)}} x_{q_i} & \text{otherwise} \end{cases}$$

$$\text{here } \sigma_{ij} = \frac{e^{s(q_i, d_j)}}{e^{s(q_i, d_j)} + \sum_{k \notin P_i} e^{s(q_i, d_k)}}, \text{where } s(q, d) = x_q^T z_d$$

It is simple to see DecoupledSoftmax is optimized when we have $\sigma_{il} = 1$ if $l \in P_i$ (for positives) and when the score for a negative $s(q_i, d_l)$ is significantly smaller than all positive scores, i.e., $s(q_i, d_j)$ for $j \in P_i$. Thus, DecoupledSoftmax avoids penalizing positives that are confidently predicted by the model and allows the training to better handle the imbalanced multi-label nature of XMC datasets.

We empirically validate the efficacy of our proposed modification in addressing the aforementioned issues. To achieve this, we construct a synthetic dataset that accentuates the challenges and demonstrate how our modification mitigates these challenges, in contrast to its absence. Specifically, we create this synthetic dataset as follows: We randomly sample 1000 training query texts and 5000 label texts. For the initial 100 training queries, we intentionally designate the first token as "t*" (a randomly selected token) and associate all these queries with the first five labels i.e. labels 0, 1, 2, 3, and 4. However, for the 0th label, we append the token "t*" to its label text. This augmentation aims to facilitate the model's learning of the pattern that whenever "t*" is present in a query, label 0 should

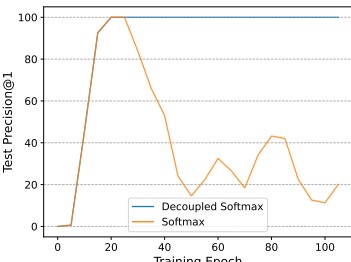

Figure 2: Decoupled softmax vs standard softmax on synthetic dataset.

be confidently predicted. The test set comprises 1000 new, randomly generated query texts, wherein we again replace the first token with "t*" and exclusively tag these queries with the 0th label.

As illustrated in Figure 2, our proposed modification results in a 100% $P@1$ on this dataset, while the standard softmax approach achieves approximately 20% $P@1$. This stark contrast highlights the softmax's inability to capture this straightforward correlation.

We note that similar situations arise in real-world XMC datasets. For instance in EURLex-4K dataset, labels such as 'afghanistan,' 'international_sanctions,' 'foreign_capital,' and 'cfsp' frequently co-occur in queries. However, 'afghanistan' is easily predictable, as training queries consistently contain the 'afghanistan' keyword, whereas the predictability of the other labels is not trivial. Thus, the observations in Figure 2 are not limited to artificially crafted scenarios, but extend to practical XMC datasets, emphasizing the relevance of our proposed modification. Figure 3 visualizes the gradients for two labels from EURLex-4K dataset for all positive training queries encountered during training. This plot reveals that regular softmax loss gives very high variance gradient feedback, but decoupled softmax gives much more consistent gradient feedback and positive feedback.

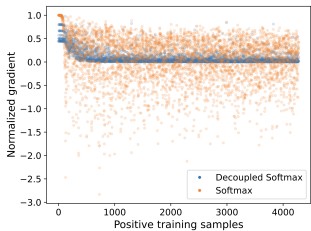 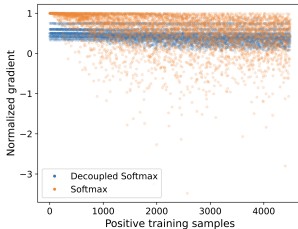

Figure 3: Gradient analysis of two labels [*left*] "afghanistan" (cherry-picked) and [*right*]"data_transmission" (randomly-chosen) on EURLex-4K dataset for all positive training queries encountered during training.

## 4.2 MEMORY EFFICIENT TRAINING

Training a DE model with the above loss function for a very large number of labels is challenging. As the pool of labels to be considered in the loss computation grows, the computational requirements to process these examples increase significantly – as for each query and label, the model needs to compute the embeddings and store all the intermediate activations to facilitate backpropagation. In order to overcome these memory constraints, we utilize a modified version of gradient cache approach (Gao et al., 2021) to perform distributed DE training with a large pool of labels under limited GPU memory. Due to space constraints, we describe our implementation and its runtime/memory analysis in detail in Section B.5 of the appendix.

Although, with the memory-efficient implementation we are able to train DE with the full loss even on the largest XMC benchmarks and establish the capabilities of DE models for XMC tasks with a modest GPU setup (see Table 9), the training is still expensive and is not desirable for scaling to even bigger datasets. To this end, we explore (see Section 5.4) approximations of our proposed approach with standard negative mining techniques similar to Xiong et al. (2021) and evaluate the performance of the learned DE model with the increasing number of negative samples. Furthermore, in Section C.3, following the embedding cache approach proposed in Lindgren et al. (2021), we also study how the quality of the sampled negatives affects the model performance.

## 4.3 PROPOSED DIFFERENTIABLE TOP-K OPERATOR-BASED LOSS

Many-shot retrieval systems are applicable in a variety of fixed-size retrieval settings like say product-product retrieval – similar to LF-AmazonTitles-1.3M dataset – where the goal is to retrieve top-k products for a given input product. While our modification to InfoNCE loss does tend to give higher scores to positives than the negatives, it might be a bit too harsh in certain cases. For example, consider there are five positives and it is difficult to rank all the five positives over all the negatives. Suppose the goal is to optimize Recall@10. Then as long as the five positives are ranked above *all but five* negatives, the loss should be small (ideally zero). However, the proposed loss in (5) would still penalize such solutions.

To explicitly optimize such top-k metrics, we propose to use a differentiable soft top-k operator to define the loss function. The soft top-k operator takes a score vector as input, and essentially serves as a filter, assigning values close to 1 for the top-k scores and values close to 0 for the remaining scores. It is designed to be differentiable to allow gradients to flow through it during backpropagation.

Once we have the soft top-k score vector, we apply a log-likelihood loss over this vector. This loss serves to provide the model with feedback on its performance in placing all positives among the top-K predictions. When all positive labels are among the top-k predictions, the loss is zero. However, if any positive label falls outside the top-k scores, the loss provides a non-zero feedback signal that guides the model to adjust its parameters during backpropagation. That is, given query $q_i$, let the score vector $\mathbf{s}_i := \big( s(q_i, d_1), \ldots, s(q_i, d_L) \big) \in \mathbb{R}^L$ and $\mathbf{z}_i := \mathsf{SoftTop\text{-}k}(\mathbf{s}_i)$ be the output of the soft top-k operator. With $\mathbf{y}_i \in \{0,1\}^L$ as the label vector, the loss function for $q_i$ takes the form:[1]

$$(\mathsf{SoftTop\text{-}k}) \quad \ell(q_i, \mathbf{y}_i; s) = -\frac{1}{L} \sum_{j \in [L]} y_{ij} \cdot \log z_{ij}. \tag{6}$$

Since $z_{ij}$ depends on the full score vector $\mathbf{s}_i$, this loss also provides feedback to all the positives and negatives in the dataset. To complete the loss function formulation, we specify the differentiable soft top-k operator below. Note that the operator is based on a proposal by Ahle (2022).

**Soft differentiable top-k formulation.** We first look at a variational definition of hard top-k operator. Let $\mathbf{x} = \{x_i\}_{i=1}^N$ be the input vector, then $\mathsf{HardTop\text{-}k}(x) = \{s(x_i + t_{\mathbf{x}})\}_{i=1}^N$, where $s(a) = I[a > 0]$ is the step function that outputs 1 if $a > 0$ and 0, otherwise. $t_{\mathbf{x}}$ is a threshold so that exactly $k$ elements exceed it, i.e., $\sum_{i=1}^N s(x_i + t_{\mathbf{x}}) = k$; for simplicity, let $\mathbf{x}$ comprise unique elements.

Based on HardTop-k operator, we can now define SoftTop-k operator by essentially replacing the above hard step function with a smooth $\sigma$ function such as the standard sigmoid function. We can change the smoothness of our soft approximator by changing the $\sigma$ function. We use a scaled sigmoid function i.e. $\sigma(x) = \mathrm{sigmoid}(\alpha x)$ where $\alpha > 0$ is a hyperparameter that controls the smoothness. In general, we might not have a closed-form solution for $t_{\mathbf{x}}$ depending on our choice of $\sigma$. However, for monotonic $\sigma$, we can find $t_{\mathbf{x}}$ using binary search: start with an under- and over-estimate of $t_{\mathbf{x}}$ and at every iteration reduce the search space by half based on comparison of $\sum \sigma(x_i + t_{\mathbf{x}})$ and $k$.

**Computing gradient of SoftTop-k operator.** By definition, $\mathsf{SoftTop\text{-}k}(\mathbf{x}) = \{\sigma(x_i + t_{\mathbf{x}})\}_{i=1}^N$, where $t_{\mathbf{x}}$ is computed using binary search. Now,

$$\sum_j \sigma(x_j + t_{\mathbf{x}}) = k \implies \frac{\partial t_{\mathbf{x}}}{\partial x_i} = \frac{-\sigma'(x_i + t_{\mathbf{x}})}{\sum_j \sigma'(x_j + t_{\mathbf{x}})}.$$

Hence, $\dfrac{\partial \mathsf{SoftTop\text{-}k}(\mathbf{x}_i)}{\partial x_l} = \sigma'(x_i + t_{\mathbf{x}})(\mathbb{1}_{[i=l]} + \dfrac{\partial t_{\mathbf{x}}}{\partial x_l}) = \sigma'(x_i + t_{\mathbf{x}})(\mathbb{1}_{[i=l]} - \dfrac{\sigma'(x_l + t_{\mathbf{x}})}{\sum_j \sigma'(x_j + t_{\mathbf{x}})}).$ (7)

Combining definition of SoftTop-k operator along with associated loss function in (6), and by using gradient of SoftTop-k in (7), we now get an approach – SoftTop-k DE – to train DE models in extreme multi-label kind of scenarios. Note that here again we have to compute gradient wrt all negatives; thus, requiring the memory efficient and distributed backpropagation implementation from Section 4.2.

## 5 EXPERIMENTS

In this section we compare DE models trained with the proposed loss functions to XMC methods. We show that the proposed loss functions indeed substantially improve the performance of DE models, even improving over SOTA in some settings. We also present ablations validating choices such as loss functions, number of negatives, and further provide a more exhaustive analysis in Section C of the Appendix

### 5.1 DATASETS AND EVALUATION

We experiment with diverse XMC datasets, EURLex-4K, LF-AmazonTitles-131K, LF-Wikipedia-500K, and LF-AmazonTitles-1.3M, each containing label texts. These datasets cover various applications like product recommendation, Wikipedia category prediction, and EU-law document tagging.

---

[1]Note that, with slight abuse of notation, we use SoftTop-k($\cdot$) and SoftTop-k to refer to a soft version of the top-k operator and the corresponding loss function, respectively.

Table 1: Performance comparison on large XMC benchmark datasets. Methods suffixed with ×3 use an ensemble of 3 learners. Bold entries represent overall best numbers while underlined entries represent best numbers among pure DE methods. Results for most existing XMC baselines are from either their respective papers or the XMC repository (Bhatia et al., 2016), blank entries indicate source does not have those numbers.

| Method | Label Text | Label Classifier | P@1 | P@5 | N@5 | PSP@5 | P@1 | P@5 | N@5 | PSP@5 |
|---|---|---|---|---|---|---|---|---|---|---|
| | | | LF-Wikipedia-500K | | | | LF-AmazonTitles-1.3M | | | |
| DistilBERT OvA Classifier | ✗ | ✓ | 82.00 | 48.54 | 73.44 | 48.87 | 48.72 | 39.09 | 44.79 | 25.91 |
| ELIAS | ✗ | ✓ | 81.96 | 48.90 | 73.71 | 48.92 | 49.26 | 39.29 | 45.44 | 27.37 |
| XR-Transformer ×3 | ✗ | ✓ | 81.62 | 47.85 | 72.43 | 47.81 | 50.14 | 39.98 | 46.59 | 27.79 |
| ECLARE ×3 | ✓ | ✓ | 68.04 | 35.74 | 56.37 | 38.29 | 50.14 | 40.00 | 46.68 | 30.56 |
| SiameseXML ×3 | ✓ | ✓ | 67.26 | 33.73 | 54.29 | 37.07 | 49.02 | 38.52 | 45.15 | 32.52 |
| NGAME | ✓ | ✓ | 84.01 | 49.97 | 75.97 | 57.04 | 56.75 | 44.09 | 52.41 | 35.36 |
| NGAME DE | ✓ | ✗ | 77.92 | 40.95 | - | 57.33 | 45.82 | 35.48 | - | 36.80 |
| DEXA | ✓ | ✓ | 84.92 | 50.52 | 76.80 | 58.33 | 56.63 | 43.90 | 52.37 | 34.86 |
| DEXA DE | ✓ | ✗ | 79.99 | 42.52 | - | 57.68 | 51.92 | 38.86 | - | **37.31** |
| DecoupledSoftmax DEXML | ✓ | ✗ | **85.78** | 50.53 | **77.11** | 58.97 | **58.40** | **45.46** | **54.30** | 36.58 |
| SoftTop-5 DEXML | ✓ | ✗ | 82.93 | **51.11** | 76.61 | **59.55** | 54.41 | 43.82 | 51.70 | 32.20 |

We follow standard setup guidelines from the XMC repository (Bhatia et al., 2016) for LF-* datasets. For EURLex-4K, we adopt the XR-Transformer setup due to unavailable raw texts in the XMC repository version. Dataset statistics are in Table 6, and a summary is in Table 5. Model evaluation employs standard XMC and retrieval metrics (P@1, P@5, PSP@5, nDCG@5, R@10, R@100). Details are in Section B.4.

## 5.2 BASELINES AND SETUP

We compare our approach with SOTA XMC methods such as DEXA (Dahiya et al., 2023b), NGAME (Dahiya et al., 2023a), XR-Transformer (Zhang et al., 2021), ELIAS (Gupta et al., 2022), ECLARE (Mittal et al., 2021b), SiameseXML (Dahiya et al., 2021). To keep the comparison fair, we use the same 66M parameter `distilbert-base` transformer (Sanh et al., 2019) as used in NGAME in all of our dual-encoder experiments and share the encoder parameters for both query and document. We also compare against encoder only version of NGAME and DEXA (represented by NGAME DE and DEXA DE), i.e., retrieval using only encoder representation. In order to have a direct comparison between a pure classification model trained with all negatives and our proposed approach, similar to the BERT-OvA approach in Gupta et al. (2022, Table 2), we also report results for the Distil-BERT OvA classifier model which stacks a classification layer with $L$ labels on top of a `distilbert-base` transformer and fine-tunes both the classification layer and the transformer by using the full one-vs-all BCE loss in (2). See Appendix B for more details on the experimental setup.

## 5.3 COMPARISON WITH XMC METHODS

Table 1 presents the main comparison results, showcasing the performance of our approach relative to existing methods. On large-scale datasets such as LF-Wikipedia-500K and LF-AmazonTitles-1.3M, our approach significantly outperforms all existing XMC methods. More specifically, it can be upto 2% better than the next best method on P@K and up to 2.5% better on PSP@k while being $20\times$ smaller (in terms of trainable parameters) than existing SOTA method on LF-AmazonTitles-1.3M (cf. Figure 1). Moreover, compared to the DE versions of NGAME and DEXA our approach can be 6% better at P@k.

On smaller datasets such as EURLex-4K, XR-Transformer outperforms our approach but as we demonstrate in Section C, its gains are mainly because it uses an ensemble of 3 learners and sparse classifier-based ranker. Such add-ons can be incorporated into any method. On LF-AmazonTitles-131K, our approach is comparable to DEXA/NGAME on P@5 but there is a considerable gap on P@1, which can be partially explained by a significant jump in performance at P@1 achieved by using a label propensity aware score fusion module in NGAME; see Dahiya et al. (2023a, Table 8).

## 5.4 DecoupledSoftmax WITH HARD NEGATIVES

Since our proposed DecoupledSoftmax loss uses all negatives in its formulation, we investigate approximation of this loss by sampling only a few negatives per train query in the batch; see Table 3.

Table 2: Performance comparison on small extreme classification datasets. Please refer to Table 1 for notations.

| Method | Label Text | Label Classifier | P@1 | P@5 | N@5 | PSP@5 | P@1 | P@5 | N@5 | PSP@5 |
|---|---|---|---|---|---|---|---|---|---|---|
| | | | EURLex-4K | | | | LF-AmazonTitles-131K | | | |
| DistilBERT OvA Classifier | ✗ | ✓ | 85.25 | 59.96 | 69.83 | 53.72 | 37.55 | 18.49 | 40.75 | 40.18 |
| ELIAS | ✗ | ✓ | 86.88 | 61.73 | **71.65** | 55.62 | 39.26 | 19.02 | 42.23 | 41.89 |
| XR-Transformer×3 | ✗ | ✓ | **88.41** | **63.18** | - | - | 38.10 | 18.32 | 40.71 | 39.59 |
| NGAME | ✓ | ✓ | - | - | - | - | 46.01 | 21.47 | 48.67 | 49.43 |
| NGAME DE | ✓ | ✗ | - | - | - | - | 42.61 | 20.69 | - | 48.71 |
| DEXA | ✓ | ✓ | - | - | - | - | **46.42** | **21.59** | **49.00** | **49.65** |
| DEXA DE | ✓ | ✗ | - | - | - | - | 44.76 | 21.18 | - | 49.50 |
| DecoupledSoftmax DEXML | ✓ | ✗ | 86.78 | 60.19 | 70.37 | 54.78 | 42.52 | 20.64 | 46.33 | 47.40 |
| SoftTop-5 DEXML | ✓ | ✗ | 83.42 | 60.95 | 70.07 | **56.84** | 41.11 | 21.23 | 46.58 | 48.70 |

Following Xiong et al. (2021), for mining the hard negatives we compute the top 100 negative shortlist at regular intervals and sample the negatives from this shortlist. For each query we train on the full pool of documents sampled in the mini-batch; as a result, for a mini-batch of size $B$ and number of hard negatives sampled per query $m$, we compute the loss on almost $B(1 + m)$ documents. Clearly from Table 3, increasing the number of effective labels considered in the loss significantly increases the performance of the approximate versions. In Section C.2.3 of the appendix we also compare the results of InfoNCE loss with hard negatives and DecoupledSoftmax loss with hard negatives.

Table 3: Ablation of number of negatives sampled per query in the DecoupledSoftmax DEXML training.

| Batch size | Hard neg per query | Effective doc pool | P@1 | P@5 | R@10 | R@100 |
|---|---|---|---|---|---|---|
| | | LF-AmazonTitles-1.3M | | | | |
| 8192 | 0 | $\sim 8192$ | 42.15 | 32.97 | 29.28 | 57.48 |
| 8192 | 1 | $\sim 16384$ | 49.16 | 39.07 | 32.76 | 60.09 |
| 8192 | 2 | $\sim 24576$ | 50.74 | 40.14 | 33.31 | 60.45 |
| 8192 | 5 | $\sim 49152$ | 52.04 | 40.74 | 33.48 | 60.13 |
| 8192 | 10 | $\sim 90112$ | 54.01 | 42.08 | 34.19 | 61.04 |
| | | LF-Wikipedia-500K | | | | |
| 2048 | 0 | $\sim 2048$ | 77.71 | 43.32 | 69.24 | 88.12 |
| 2048 | 1 | $\sim 4096$ | 82.85 | 48.84 | 74.47 | 90.08 |
| 2048 | 2 | $\sim 6144$ | 83.34 | 49.32 | 74.73 | 89.74 |
| 2048 | 5 | $\sim 12288$ | 83.86 | 49.57 | 74.60 | 89.12 |
| 2048 | 10 | $\sim 22528$ | 84.77 | 50.31 | 75.52 | 90.29 |

## 5.5 Comparison across different loss functions

Next, we compare different loss formulations as discussed in Sections 3 and 4 on the EURLex-4K and LF-AmazonTitles-131K datasets (cf. Table 4). The DE model fails to train with BCE loss in (2). We hypothesize that this might be due to the stringent demand of the BCE loss, which requires all negative pairs to exhibit high negative similarity and all positive pairs to have a high positive similarity – a feat that can be challenging for DE representations. DecoupledSoftmax loss in (5) performs the best for top-1 predictions. Moreover, the SoftTop-5 and SoftTop-100 losses (cf. (6)) yield optimal results for top-5 and top-100 predictions, respectively. These results show that SoftTop-k loss can outperform other loss formulations when optimizing for a specific prediction set size. We further analyze the label score distributions of models learned with different loss functions in Section C of the Appendix.

Table 4: Ablation of DEXML loss functions

| Loss | P@1 | P@5 | R@10 | R@100 |
|---|---|---|---|---|
| | EURLex-4K | | | |
| BCE (2) | 0.1 | 0.07 | 0.14 | 1.84 |
| SoftmaxCE (3) | 80.05 | 58.36 | 72.41 | 92.57 |
| DecoupledSoftmax (5) | **86.78** | 60.19 | 72.56 | 91.75 |
| SoftTop-5 (6) | 83.42 | **60.95** | **74.21** | 91.30 |
| SoftTop-100 (6) | 52.34 | 37.41 | 56.97 | **93.72** |
| | LF-AmazonTitles-131K | | | |
| BCE (2) | 12.75 | 5.96 | 17.18 | 25.20 |
| SoftmaxCE (3) | 41.77 | 20.87 | 56.91 | 69.49 |
| DecoupledSoftmax (5) | **42.52** | 20.64 | 56.36 | 68.52 |
| SoftTop-5 (6) | 41.11 | **21.23** | **57.72** | 69.46 |
| SoftTop-100 (6) | 33.38 | 18.27 | 53.87 | **71.52** |

## 6 Conclusions & Limitations

Our key contribution is a simple yet effective training procedure for DE methods that allows them to match or surpass SOTA extreme classification methods on standard benchmarks for many-shot retrieval. One limitation of our empirical evaluation is that we have only evaluated our proposal up to O(million)-sized label spaces. Verifying our gains on O(billion) scale is an interesting and challenging avenue for future work. Additionally, designing algorithms that can improve DE performance on XMC tasks without requiring all the negatives is another interesting direction. We investigated a well known many-shot learning problem and studied it in abstract form, so we don't envision significant additional negative implications for society.

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

Table 5: A snapshot of the datasets used in our experiments. Each row represents a different dataset with a sample query and associated labels/documents separated by ";"

| Dataset | Query | Labels/Documents |
|---|---|---|
| EURLex-4K | "commiss decis octob aid scheme implement kingdom spain airlin intermediación aérea sl notifi document number spanish text authent..." | air_transport; airline; catalonia control_of_state_aid; invitation_to_tender; services_of_general_interest; spain; state_aidtransport_company |
| LF-AmazonTitles-131K | "Snowboard Kids 2" | Super Mario 64; Super Smash Bros.; Nintendo 64 System - Video Game Console; Donkey Kong 64; Diddy Kong Racing; Kirby 64: The Crystal Shards |
| LF-Wikipedia-500K | "bill slavick is an american retired professor and peace activist who ran for the u s senate in maine as an independent politician independent in the maine..." | 1928 births; american roman catholics; living people; louisiana state university faculty; maine independents; politicians from portland maine; university of notre dame alumni |
| LF-AmazonTitles-1.3M | "Power Crunch 12-1.4 oz Cookies Peanut Butter Fudge Energy Bars BNRG" | RiseBar Protein Almond Honey, 2.1 oz. Bars, 12-Count; thinkThin Protein Bars, Caramel Fudge, Gluten Free, 2.1-Ounce Bars (Pack of 10); Vitafusion Fiber Gummies Weight Management, 90-Count Bottle; Kirkland Signature Extra Fancy Unsalted Mixed Nuts 2.5 (LB); 24 Detour Bars, Low Sugar Whey Protein Bars, 1.5oz Bars; BioNutritional Research Group Power Crunch Protein Triple Ch; Dymatize ISO100 Hydrolyzed 100% Whey Protein Isolate Gourmet Vanilla - 5 lbs; thinkThin Chocolate Espresso Gluten Free, 2.1-Ounce Bars (Pack of 10)... |

## A  FUTURE WORK AND REPRODUCIBILITY

While our work shows that dual encoder model can be better or as competent as SOTA extreme classification methods without needing to linearly scale the model parameters with the number of labels, there are still challenges that need to be addressed. Below we highlight a few key future work directions:

- **Scaling to larger label spaces:** We have established the performance of small dual encoders on up to O(million) label spaces. However, it would be interesting to study up to how big datasets can the limited-size dual encoders can retain their performance and if there is any fundamental capacity threshold that is essential for scaling to practical web-scale datasets.

- **Improving DE performance without all negatives:** We have observed a clear relationship between DE performance and the availability of negatives. While it is computationally challenging to have all negatives, designing algorithms that can boost DE performance on many-shot retrieval problems without requiring a large pool of documents in loss computation is another promising area for exploration.

- **DE encoder with in-built fast search procedure:** DE approaches rely on external approximate nearest neighbor search routines to perform fast inference when searching for top documents for a given query. In contrast, many hierarchical extreme classification methods come in-built with fast search procedures, it would be interesting to explore such hierarchical approaches for DE models.

**Reproducibility**: Our `PyTorch` based implementation is available at the following link. We plan to open-source the code after the review period along with the trained models to facilitate further research in these directions. Below we provide additional details to clarify our approach and experimental setup.

## B  IMPLEMENTATION DETAILS

### B.1  DATASET

In this subsection we provide more details of the datasets considered in this work,

- **EURLex-4K**: The EURLex-4K dataset is a small extreme classification benchmark dataset. It was sourced from the publications office of the European Union (EU). Specifically, it's a collection of documents concerning European Union law. The dataset comprises approximately 4,000 distinct labels, represented by EuroVoc descriptors. EuroVoc is a multilingual thesaurus maintained by the EU, covering fields of EU interest and offering a consistent set of terms for the categorization of documents. There are over 15,000 training examples in the dataset. Each document within the EURLex-4K dataset is provided as a text string, consisting of the title and the content (or body) of the document.

- **LF-AmazonTitles-131K**: Originated from Amazon, this dataset encapsulates product relationships, aiming to predict associated products based on a queried product. It comprises about 131,000 labels (products for recommendation from the catalog), and around 300,000

training query products. The products are recognized via their respective Amazon product titles, which function as the principal descriptive feature.

- **LF-Wikipedia-500K**: A conventional large-scale extreme classification benchmark, this dataset comprises document tagging of Wikipedia articles. Human annotators have tagged query documents with multiple categories, amounting to a total of 500,000 labels and approximately 1.8 million training queries. It is marked by the presence of certain super-head classes.

- **LF-AmazonTitles-1.3M**: This dataset is similar to LF-AmazonTitles-131K, albeit on a larger scale and with a denser network. It includes a total of 2.2 million training points and 1.3 million labels. The distribution of these co-purchasing links can be significantly skewed. Some popular products relate to a considerable number of other products, while less popular ones maintain fewer connections.

Table 6: Dataset statistics

| Dataset | Num Train Points | Num Test Points | Num Labels | Avg Labels per Point | Avg Points per Label |
|---|---|---|---|---|---|
| EURLex-4K | 15,449 | 3,865 | 3801 | 5.30 | 20.79 |
| LF-AmazonTitles-131K | 294,805 | 134,835 | 131,073 | 2.29 | 5.15 |
| LF-Wikipedia-500K | 1,779,881 | 769,421 | 501,070 | 4.75 | 16.86 |
| LF-AmazonTitles-1.3M | 2,248,619 | 970,237 | 1,305,265 | 22.20 | 38.24 |

## B.2 TRAINING HYPERPARAMETERS

In all of our experiments we train the dual encoders for 100 epochs with `AdamW` optimizer and use the linear decay with warmup learning rate schedule. Following the standard practices, we use $0$ weight deacy for all non-gain parameters (such as layernorm, bias parameters) and $0.01$ weight decay for all the rest of the model parameters. Rest of the hyperparameters considered in our experiments are described below:

- `max_len`: maximum length of the input to the transformer encoder, similar to (Dahiya et al., 2023a) we use $128$ `max_len` for the long-text datasets (EURLex-4K and LF-Wikipedia-500K) and $32$ `max_len` for short-text datasets (LF-AmazonTitles-131K and LF-AmazonTitles-1.3M)

- `LR`: learning rate of the model

- `batch_size`: batch-size of the mini-batches used during training

- `dropout`: dropout applied to the dual-encoder embeddings during training

- $\alpha$: multiplicative factor to control steepness of $\sigma$ function described in Section 4.3

- $\eta$: micro-batch size hyperparameter that controls how many labels get processed at a time when using gradient caching

- $\tau$: temperature hyperparameter used to scale similarity values (i.e. $s(q_i, d_j)$) during loss computation

Table 7: Hyperparameters

| Dataset | max_len | LR | batch_size | dropout | $\alpha$ | $\eta$ | $\tau$ |
|---|---|---|---|---|---|---|---|
| EURLex-4K | 128 | 0.0002 | 1024 | 0.1 | 2 | 4096 | 0.05 |
| LF-AmazonTitles-131K | 32 | 0.0004 | 4096 | 0 | 2 | 2048 | 0.05 |
| LF-Wikipedia-500K | 128 | 0.0004 | 4096 | 0.1 | 2 | 2048 | 1 |
| LF-AmazonTitles-1.3M | 32 | 0.0008 | 8192 | 0.1 | 2 | 1024 | 1 |

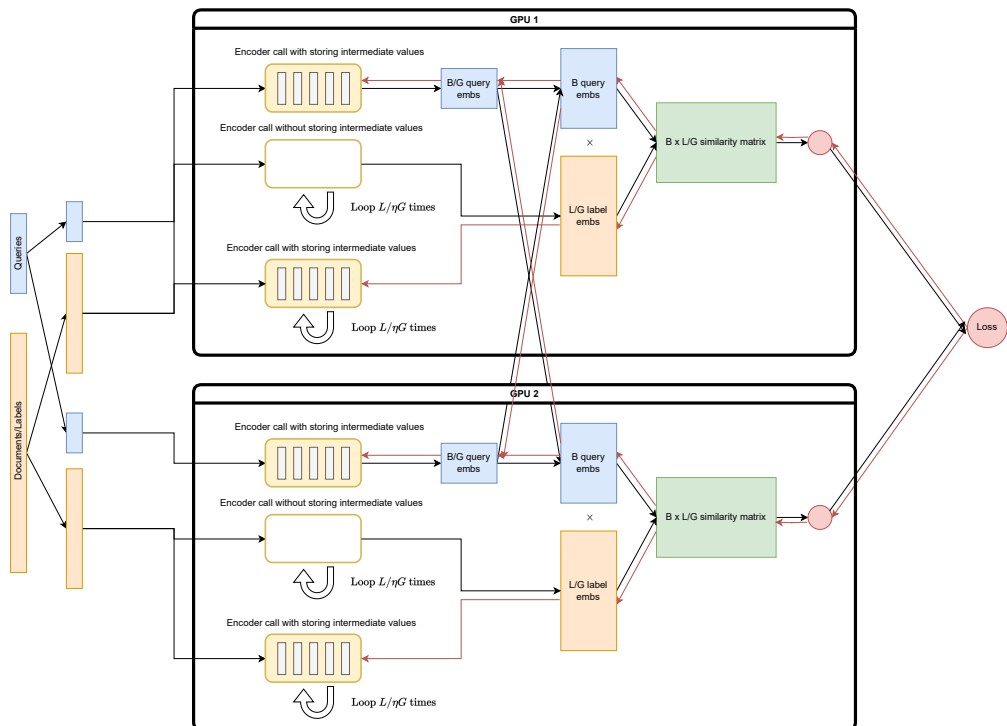

Figure 4: Illustration of distributed implementation with gradient caching applied on label embedding computation. Here solid black line indicate forward pass direction and solid red lines indicate gradient backpropagation direction. $L$ is the number of labels considered in the loss computation, $B$ is the batch size of queries, $\eta$ is a micro batch-size hyperparameter which controls how many labels are processed at a time.

### B.3 BASELINES AND EVALUATION METRICS

For existing extreme classification baselines (apart from ELIAS), we obtain their numbers from their respective papers or the extreme classification repository (Bhatia et al., 2016). Since the ELIAS paper do not report results on most of the datasets considered in this work, we ran the publicly available ELIAS implementation to report the results.

### B.4 EVALUATION METRICS

We evaluate the models using standard XMC and retrieval metrics such as P@1, P@5, PSP@5, nDCG@5, R@10, and R@100. P@k (*Precision* at k) measures the proportion of the top-k predicted labels that are also among the true labels for a given instance. PSP@k (*Propensity Scored Precision* at k) is a variant of precision that takes into account the propensity scores of the labels, which are indicative of the likelihood of a label being positive. nDCG@k (*normalized Discounted Cumulative Gain* at k) measures the quality of the ranked list of labels up to the position k. R@k (*Recall* at k) is the proportion of the true labels that are included in the top-k predicted labels.

### B.5 MEMORY EFFICIENT DISTRIBUTED IMPLEMENTATION

In this section we describe our distributed implementation in more detail. For a multi-GPU setting, where we have $G$ GPUs, a batch size of $B$ queries, and a pool size of $L$ labels considered in the loss computation.

We begin by distributing the data and labels evenly across the GPUs. Each GPU receives $B/G$ queries and $L/G$ labels. However, for extreme multi-label problems with > 1M labels, even this reduced load can be too large for the $L/G$ labels – memory-wise – for an individual GPU during backpropagation. To handle this, we use the *gradient caching* method (Gao et al., 2021). More concretely, on each GPU, we first divide the large $L/G$ label input batch into manageable sub-batches of size $\eta$. Then the encoder performs $L/\eta G$ forward passes to compute label embeddings without

constructing the computation graph (i.e. `with torch.no_grad()`). Then, the loss is computed based on these embeddings, and the gradients for each embedding are computed and stored in a cache. Once we get the gradients with respect to the label embeddings, the encoder is run again on each $\eta$ sized sub-batch separately (but this time we retain the computation graph), and the stored embedding gradients are used to backpropagate through the encoder parameters. This allows for constant memory usage during the encoder's forward-backward pass at the cost of doing the forward pass twice.

To compute the loss, we perform an *all-gather* operation on query embeddings across all GPUs. This enables each GPU to compute the dot product similarity matrix for the full $B$ queries and $L/G$ labels. The computation of the loss requires certain common terms, such as the denominator in the case of softmax-style losses. To compute these common terms, we perform an *all-reduce* operation across all GPUs. Finally, each GPU computes the gradients with respect to its batch of $L/G$ label and $B/G$ query embeddings and backpropagates the respective gradients through the encoder parameters. By computing the loss and gradients in this distributed manner and by employing gradient cache, we can effectively handle large pools of labels without overwhelming the memory capacity of individual GPUs.

In the following description we assume that the loss is standard softmax loss but appropriate modifications can be done to make it work for other losses discussed in Section 4. Figure 4 illustrates the overall processing of a mini-batch in a 2 GPU setup. More generally, let there be $G$ GPUs, $B$ queries $\{x_i\}_{i=1}^B$ and $L$ labels $\{l_i\}_{i=1}^L$ considered in the loss computation.

- Each GPU receives $B/G$ queries and $L/G$ labels. More concretely, we define the set of queries received by the $g^{th}$ GPU as $\{x_i^g\}_{i=1}^{B/G}$ and the set of labels as $\{l_i^g\}_{i=1}^{L/G}$.

- We can then define the embedding matrices for each GPU for $B/G$ queries and $L/G$ labels as $Q_g = \phi(\{x_i^g\}_{i=1}^{B/G})$ and $L_g = \phi(\{l_i^g\}_{i=1}^{L/G})$ respectively, where $\phi$ is the query/document encoder. Note that, computing $L_g$ for large $L$ requires gradient caching as described in Section 4.2.

- We then gather all query embeddings at each GPU, denoted as $\hat{Q} = \text{Gather}(Q_g)_{g=1}^G$

- We compute the similarity matrix for $B$ queries and $L/G$ labels as $S_g = \hat{Q} L_g^T$.

- Finally, we compute the softmax loss for each query-label pair $(i, j)$ (on respective GPUs) as $\ell_{i,j} = -(S_{i,j} - \texttt{logsumexp}_{j=1}^L(S_{i,j}))$. Note, since $S = [S_1, S_2, ..., S_G]$ is distributed across GPU, we will have to perform an *all_reduce* operation to compute $\texttt{logsumexp}_{j=1}^L(S_{i,j})$ term in the loss.

### B.5.1 Memory and Runtime Analysis of Proposed implementation

Below we provide analysis of memory and compute requirements of the implementation with or without gradient cache based approach: let's assume we have $G$ gpus, our batch size is $B$, number of labels $L$, gradient cache sub-batch size $\eta$, embedding dimension $d$ and the memory required for processing (combined forward and backward pass of transformer encoder during training) a single query and label is $M_q^{enc}$ and $M_l^{enc}$ respectively. Without gradient caching, the total memory requirements for processing the whole mini-batch during training will be: $\mathcal{O}(\frac{B(M_q^{enc}+d)+L(M_l^{enc}+d)+BL}{G})$, notice that with transformer encoders $LM_l^{enc}$ is the largest bottleneck for XMC datasets with $L$ in order of millions. With gradient caching the memory requirement is: $\mathcal{O}(\frac{B(M_q^{enc}+d)+Ld+\eta M_l^{enc}+BL}{G})$. Similarly for compute let's assume the compute required for the forward and backward pass of query be $T_q^{enc-f}$ and $T_q^{enc-b}$ respectively, the compute required for the forward and backward pass of a label be $T_l^{enc-f}$ and $T_l^{enc-b}$. Then, without gradient caching the total compute requirement is: $\mathcal{O}(\frac{B(T_q^{enc-f}+T_q^{enc-b}+d)+L(T_l^{enc-f}+T_l^{enc-b}+d)+BL}{G})$. With gradient caching the compute requirement will be: $\mathcal{O}(\frac{B(T_q^{enc-f}+T_q^{enc-b}+d)+L(2T_l^{enc-f}+T_l^{enc-b}+d)+BL}{G})$.

Empirically even with 16 A100 GPUs, the naive approach without gradient caching runs easily out of GPU memory even on LF-AmazonTitles-131K dataset as it demands about $\sim$TB combined GPU memory just for encoding all the labels. Table 8 provides the ablation of memory used and runtimes

of the memory-efficient approach with different global batch sizes and gradient cache sub-batch sizes on LF-AmazonTitles-131K dataset (experiments ran with 4 A100 GPU setup):

Table 8: Ablation of memory used and runtimes of the grad-cache based memory efficient implementation

| Global batch size | Gradient cache sub-batch size | Max Memory per GPU | Avg time per batch | Avg time per epoch |
|---|---|---|---|---|
| 1024 | 64 | 6.03 GB | 17.8s | 1.42 hrs |
| 1024 | 256 | 6.31 GB | 7.1s | 0.56 hrs |
| 1024 | 1024 | 14.89 GB | 5.9s | 0.47 hrs |
| 1024 | 4096 | 48.51 GB | 5.6s | 0.44 hrs |
| 4096 | 256 | 15.75 GB | 7.9s | 0.16 hrs |
| 4096 | 1024 | 16.16 GB | 6.7s | 0.14 hrs |
| 4096 | 4096 | 51.85 GB | 6.0s | 0.12 hrs |
| 16384 | 256 | 53.12 GB | 12.2s | 0.06 hrs |
| 16384 | 1024 | 53.12 GB | 8.2s | 0.04 hrs |
| 16384 | 4096 | 58.77 GB | 6.6s | 0.03 hrs |

## B.6 HARDWARE AND RUNTIMES

We run our experiments on maximum 16 A100 GPU setup each having 40 GB GPU memory. In principle, our experiments can also be performed on a single GPU setup, but it'll take significantly longer to train. Table 9 reports the training times and number of GPU used during training for each dataset.

Table 9: Training times and resources used for each dataset in our experiments

| Parameter | EURLex-4K | LF-AmazonTitles-131K | LF-Wikipedia-500K | LF-AmazonTitles-1.3M |
|---|---|---|---|---|
| Number of GPU | 2 | 4 | 8 | 16 |
| Total Training time (hrs) | 0.27 | 4.2 | 37 | 66 |

## B.7 DISTRIBUTED log SOFTTOP-$k$ IMPLEMENTATION

Below we provide the code snippet for efficient distributed implementation of $\log$ SoftTop-$k$ operator (for loss computation it is more desirable to directly work in $\log$ domain) in PyTorch, here it is assumed that xs (the input to our operator is of shape $B \times L/G$ where B is the batch size, $L$ is the total number of labels and $G$ is the number of GPUs used in the distributed setup.

```
1 # Topk code adapted from https://gist.github.com/thomasahle/4
    c1e85e5842d01b007a8d10f5fed3a18
2 sigmoid = torch.sigmoid
3 def sigmoid_grad(x):
4     sig_x = sigmoid(x)
5     return sig_x * (1 - sig_x)
6
7 from torch.autograd import Function
8 class DistLogTopK(Function):
9     @staticmethod
10    def forward(ctx, xs, k, alpha, n_iter=32):
11        logits, log_sig = _dist_find_ts(xs, k, alpha=alpha, n_iter=n_iter
    , return_log_sig=True)
12        ctx.save_for_backward(torch.tensor(alpha), logits)
13        return log_sig
14
15    @staticmethod
16    def backward(ctx, grad_output):
17        # Compute vjp, that is grad_output.T @ J.
```

```
18          alpha, logits = ctx.saved_tensors
19          sig_x = sigmoid(logits)
20          v = alpha*sig_x*(1-sig_x)
21          s = v.sum(dim=1, keepdims=True)
22          dist.all_reduce(s, op=dist.ReduceOp.SUM)
23          uv = grad_output * alpha * (1-sig_x)
24          uv_sum = uv.sum(dim=1, keepdims=True)
25          dist.all_reduce(uv_sum, op=dist.ReduceOp.SUM)
26          t1 = - uv_sum * v / s
27          return t1.add_(uv), None, None, None  # in-place addition
28
29 @torch.no_grad()
30 def _dist_find_ts(xs, k, alpha=1, n_iter=64, return_log_sig=False):
31      assert alpha > 0
32      b, n = xs.shape
33      n *= dist.get_world_size()
34      if isinstance(k, int):
35          assert 0 < k < n
36      elif isinstance(k, torch.LongTensor):
37          assert (0 < k).all() and (k < n).all()
38      # Lo should be small enough that all sigmoids are in the 0 area.
39      # Similarly Hi is large enough that all are in their 1 area.
40      xs_min = xs.min(dim=1, keepdims=True).values
41      xs_max = xs.max(dim=1, keepdims=True).values
42      dist.all_reduce(xs_min, op=dist.ReduceOp.MIN)
43      dist.all_reduce(xs_max, op=dist.ReduceOp.MAX)
44      lo = -xs_max - 10/alpha
45      hi = -xs_min + 10/alpha
46      for _ in range(n_iter):
47          mid = (hi + lo)/2
48          sigmoid_sum = sigmoid(alpha*(xs + mid)).sum(dim=1)
49          dist.all_reduce(sigmoid_sum, op=dist.ReduceOp.SUM)
50          mask = sigmoid_sum < k
51          lo[mask] = mid[mask]
52          hi[~mask] = mid[~mask]
53
54      ts = (lo + hi)/2
55      logits = alpha*(xs + ts)
56      if return_log_sig:
57          log_sig = logits - torch.logaddexp(logits, torch.zeros_like(
   logits[:, :1]))
58          return logits, log_sig
59      else:
60          return logits, sigmoid(logits)
```

## C ADDITIONAL RESULTS AND ANALYSIS

### C.1 MEMORIZATION CAPABILITIES OF DE

In order to verify if DE models are capable of memorizing random text co-relations even in the presence of an extreme semantic gap between query and label text, we create synthetic datasets of different sizes with completely random query and label texts, and then train and evaluate retrieval performance of DE models. More specifically, for a synthetic dataset of size $N$, we sample $N$ random query texts (by independently sampling 16 random tokens to form one query text) and similarly sample $N$ random label texts. We assign $i^{th}$ query to $i^{th}$ label, train the DE model with in-batch and sampled hard negatives on this dataset, and then report the retrieval performance on the same dataset. As seen in Table 10, even at a million scale DE models perform this task with perfect accuracy which shows that they are well capable of performing random text memorization.

Table 10: Performance of DE on random text pair synthetic dataset at different scales

| Dataset size ($N$) | P@1 | P@5 | R@10 | R@100 | P@1 | P@5 | R@10 | R@100 |
|---|---|---|---|---|---|---|---|---|
| | DE with in-batch negative | | | | DE with hard negative | | | |
| 100K | 100.00 | 20.00 | 100.00 | 100.00 | 100.00 | 20.00 | 100.00 | 100.00 |
| 500K | 98.60 | 20.00 | 100.00 | 100.00 | 100.00 | 20.00 | 100.00 | 100.00 |
| 1M | 81.21 | 18.67 | 95.71 | 99.15 | 99.93 | 20.00 | 99.99 | 99.99 |

## C.2 DecoupledSoftmax VS STANDARD SOFTMAX

### C.2.1 ANALYTICAL GRADIENT ANALYSIS

**Softmax gradient analysis**:

$$\mathcal{L} = -\frac{1}{N} \sum_{i=1}^{N} \ell_i$$

$$\ell_i = \sum_{j \in P_i} \ell_{ij}, \text{where } P_i \text{ is the set of positive documents of label } i$$

$$\ell_{ij} = \log(\sigma_{ij}), \text{here } \sigma_{ij} = \frac{e^{s(q_i, d_j)}}{\sum_k e^{s(q_i, d_k)}}, \text{where } s(q, d) = x_q^T z_d$$

$$\implies \frac{\partial \ell_{ij}}{\partial z_{d_l}} = \begin{cases} (1 - \sigma_{il}) x_{q_i} & \text{if } l = j \\ -\sigma_{il} x_{q_i} & \text{otherwise} \end{cases}$$

$$\implies \frac{\partial \ell_i}{\partial z_{d_l}} = \begin{cases} (1 - |P_i| \sigma_{il}) x_{q_i} & \text{if } l \in P_i \\ -|P_i| \sigma_{il} x_{q_i} & \text{otherwise} \end{cases}$$

This suggests that the multi-label extension of standard softmax is optimized when:

$$\sigma_{il} = \begin{cases} \frac{1}{|P_i|} & \text{if } l \in P_i \\ 0 & \text{otherwise} \end{cases}$$

DecoupledSoftmax **gradient analysis**:

$$\mathcal{L} = -\frac{1}{N} \sum_{i=1}^{N} \ell_i$$

$$\ell_i = \sum_{j \in P_i} \ell_{ij}, \text{where } P_i \text{ is the set of positive documents of label } i$$

$$\ell_{ij} = \log(\sigma_{ij}), \text{here } \sigma_{ij} = \frac{e^{s(q_i, d_j)}}{e^{s(q_i, d_j)} + \sum_{k \notin P_i} e^{s(q_i, d_k)}}, \text{where } s(q, d) = x_q^T z_d$$

$$\frac{\partial \ell_{ij}}{\partial z_{d_l}} = \begin{cases} (1 - \sigma_{il}) x_{q_i} & \text{if } l = j \\ 0 & \text{if } l \neq j \text{ and } l \in P_i \\ -\frac{e^{s(q_i, d_l)}}{e^{s(q_i, d_j)} + \sum_{k \notin P_i} e^{s(q_i, d_k)}} x_{q_i} & \text{otherwise} \end{cases}$$

$$\implies \frac{\partial \ell_i}{\partial z_{d_l}} = \begin{cases} (1 - \sigma_{il}) x_{q_i} & \text{if } l \in P_i \\ -\sum_{j \in P_i} \frac{e^{s(q_i, d_l)}}{e^{s(q_i, d_j)} + \sum_{k \notin P_i} e^{s(q_i, d_k)}} x_{q_i} & \text{otherwise} \end{cases}$$

### C.2.2 EMPIRICAL RESULTS ON XMC BENCHMARKS

Table 11 compares DecoupledSoftmax with SoftmaxCE loss across all XMC datasets considered in this work.

Table 11: Side by side comparison of DE models trained with DecoupledSoftmax vs Softmax loss on XMC datasets.

| Dataset | P@1 | | P@5 | |
|---|---|---|---|---|
| | Decoupled Softmax | Softmax | Decoupled Softmax | Softmax |
| EURLex-4K | 86.78 | 80.05 | 60.19 | 58.36 |
| LF-AmazonTitles-131K | 42.52 | 41.77 | 20.64 | 20.87 |
| LF-Wikipedia-500K | 85.78 | 79.85 | 50.53 | 49.78 |
| LF-AmazonTitles-1.3M | 58.40 | 52.42 | 45.46 | 43.43 |

### C.2.3 COMPARING DecoupledSoftmax AND SOFTMAX WITH HARD NEGATIVES

Table 12 compares approximations of DecoupledSoftmax and SoftmaxCE loss with hard negatives on the largest LF-AmazonTitles-1.3M dataset.

Table 12: Performance comparison of DecoupledSoftmax vs Softmax with sampled hard negatives, note that Softmax with sampled hard negatives is same as InfoNCE with hard negatives

| Batch size | Hard neg per query | Effective doc pool | P@1 | P@5 | R@10 | R@100 | P@1 | P@5 | R@10 | R@100 |
|---|---|---|---|---|---|---|---|---|---|---|
| | | | | Decoupled | Softmax | | | Softmax | | |
| 8192 | 1 | $\sim 16384$ | 49.16 | 39.07 | 32.76 | 60.09 | 47.60 | 38.85 | 32.12 | 61.49 |
| 8192 | 2 | $\sim 24576$ | 50.74 | 40.14 | 33.31 | 60.45 | 49.11 | 40.16 | 32.77 | 62.05 |
| 8192 | 10 | $\sim 90112$ | 54.01 | 42.08 | 34.19 | 61.04 | 51.19 | 42.06 | 33.31 | 62.76 |
| 8192 | All | $\sim 1.3M$ | 58.40 | 45.46 | 36.49 | 64.25 | 52.42 | 43.42 | 34.00 | 65.84 |

### C.3 QUALITY OF NEGATIVES IN DE TRAINING

As mentioned in Section 5.4 we perform negative sampling following (Xiong et al., 2021), we mine hard negatives by computing the shortlist of nearest indexed labels for each training query at regular intervals and sample negatives from this shortlist. Although, this approach is computationally efficient it suffers from the problem of having to deal with stale negatives since the model parameters constantly change but the negative shortlist gets updated only after certain training steps. To study the importance of the quality of hard negatives we implement an approach similar to (Lindgren et al., 2021) where we maintain an active cache of all label embeddings and use this to first identify the hardest negatives and then use these negatives along with in-batch labels to perform loss computation. More specifically, we maintain a label embedding cache of $L \times d$ size where $L$ is the number of labels and $d$ is the embedding dimension. We initialize this matrix with the embeddings of all the labels at the start of training. For a mini-batch of $B$ queries and $L'$ sampled labels in the mini-batch, we first update the embedding cache of all $L'$ mini-batch labels, then we compute the loss using embeddings of $B$ training queries and the label embedding cache and identify the labels that receive the highest gradients. We take the top-$K$ labels that are not already part of the $L'$ mini-batch labels. Then we use these top-k labels as the hard negative for the mini-batch and then compute the loss using the $L'$ mini-batch labels and $K$ hard negative labels. This approach allows us to mine *almost* fresh hard negatives for each mini-batch. In Table 13 we compare the results obtained using this approach and ANCE (Xiong et al., 2021) style hard negatives. Note that for a fair comparison, we compare them in settings where an equivalent total number of negatives are sampled per batch. These results suggest that there can be moderate gains made with an increase in the quality of negatives and further research into mining better negatives can be helpful in training better DE models with lower cost.

### C.4 RESULTS WITH CONFIDENCE INTERVALS ON SMALL DATASETS

Since XMC datasets are fairly big benchmarks so running multiple experiments with different seeds is very expensive. Below in Table 14 we provide mean and 95% confidence intervals for DecoupledSoftmax DE results for the smaller EURLex-4K and LF-AmazonTitles-131K datasets.

Table 13: Comparing DecoupledSoftmax DE trained with ANCE (Xiong et al., 2021) style stale negatives and fresh negatives mined using embedding cache (Lindgren et al., 2021)

| Dataset | P@1 | P@5 | R@10 | R@100 | P@1 | P@5 | R@10 | R@100 |
|---|---|---|---|---|---|---|---|---|
| | ANCE (Xiong et al., 2021) | | | | Embedding cache (Lindgren et al., 2021) | | | |
| EURLex-4K | 84.61 | 59.15 | 71.08 | 90.21 | 85.87 | 59.93 | 71.54 | 90.34 |
| LF-AmazonTitles-131K | 40.99 | 20.59 | 57.20 | 70.08 | 41.72 | 20.25 | 45.54 | 65.54 |
| LF-Wikipedia-500K | 82.85 | 48.84 | 74.47 | 90.08 | 84.53 | 48.66 | 72.97 | 88.93 |

Table 14: Results with 95% confidence intervals for DecoupledSoftmax DE trained on EURLex-4K and LF-AmazonTitles-131K dataset

| P@1 | P@5 | R@10 | R@100 | P@1 | P@5 | R@10 | R@100 |
|---|---|---|---|---|---|---|---|
| EURLex-4K | | | | LF-AmazonTitles-131K | | | |
| $86.50 \pm 0.24$ | $60.24 \pm 0.19$ | $72.62 \pm 0.26$ | $91.76 \pm 0.10$ | $42.37 \pm 0.13$ | $20.57 \pm 0.05$ | $56.25 \pm 0.09$ | $68.51 \pm 0.04$ |

## C.5 SCORE DISTRIBUTION ANALYSIS

In Figure 5 we plot the precision-recall curve on EURLex-4K for DE models trained using DecoupledSoftmax and Softmax loss. This plot reveals that DE model trained with DecoupledSoftmax offers a better precision vs recall tradeoff. In Figure 6 we analyze the distribution of scores given by DE models trained with different loss function on the test set of EURLex-4K dataset. More specifically, we plot the distribution of scores of positive and negative labels for Softmax loss 3, DecoupledSoftmax loss 5, and SoftTop-5 loss 6. These plots reveal that the positives and negatives are better separated when we use DecoupledSoftmax and SoftTop-5 loss, when compared to the Softmax loss.

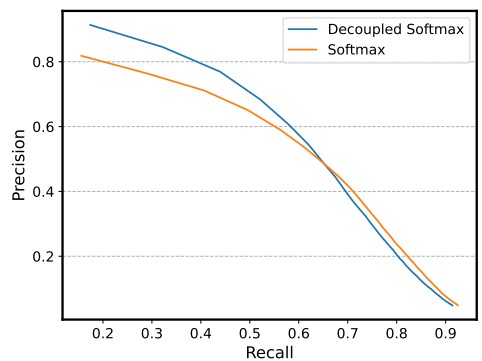

Figure 5: PR curve for Decoupled softmax vs standard softmax on EURLex-4K dataset

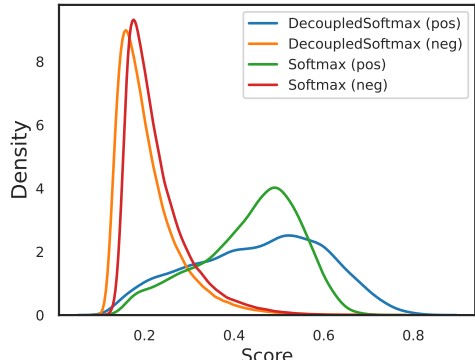
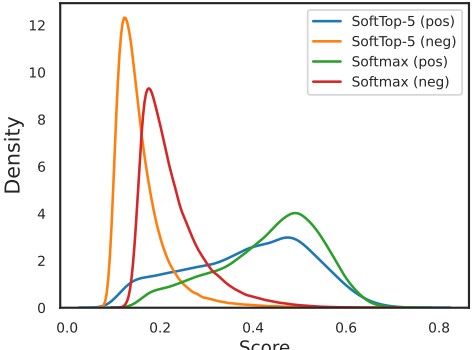

Figure 6: Plot of score distribution of DE models trained with different loss functions. *left* plot compares DecoupledSoftmax 5 vs softmax 3 *right* plot compares SoftTop-5 6 vs softmax 3

## C.6 DECILEWISE COMPARISON

We investigate the performance variation across different label regimes (many-shot versus few-shot labels) among various approaches. These include the pure classification-based method (DistilBERT OvA Classifier), DecoupledSoftmax DE models trained with in-batch negatives (In-batch DE), hard negatives (Hard negative DE), and all negatives (DecoupledSoftmax DE and SoftTop-k DE).

Consistent with Dahiya et al. (2023a), we devise different label deciles based on the count of training examples per label, wherein the first decile signifies labels with the most training examples and the last decile signifies those with the least. On LF-Wikipedia-500K, as anticipated, Figure 7 shows that the pure classification-based approach achieves subpar performance on the last deciles. However, intriguingly, DE methods trained with hard or all negatives exhibit performance comparable to the classification-based approaches even on head deciles. DE methods trained with our approach does well across *all* deciles and hence improve the performance over existing DE methods. On LF-AmazonTitles-1.3M, when comparing the DistilBERT OvA classifier with DE models, we notice a similar trend that the pure classification-based approach and DE approach are comparable at head deciles but on tail deciles DE starts significantly outperforming the classifier approach. When comparing DE trained with all negatives and the ones trained with in-batch or hard-negatives, we notice a substantial gap on head deciles but the gap diminishes on tail deciles.

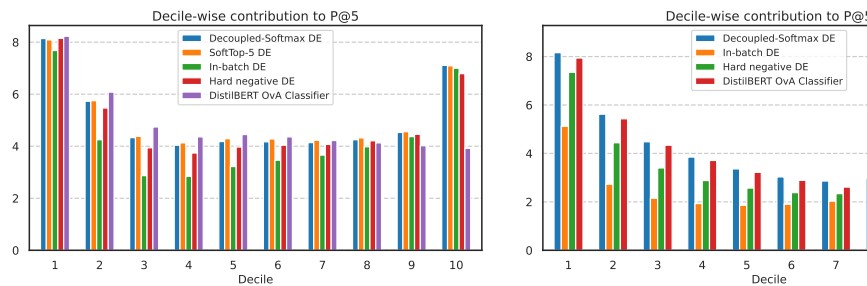

Figure 7: Decile-wise analysis of P@5 on LF-Wikipedia-500K (*left*) and LF-AmazonTitles-1.3M (*right*), $1^{st}$ decile represents labels with most training points i.e. head labels while $10^{th}$ decile represents labels with least training points i.e. tail labels

## C.7 BETTER GENERALIZATION OF DE MODELS ON TAIL LABELS

In this section, we study whether DE models offer improved learning when dealing with similar but distinct labels when compared to the classifier approach. We use LF-Wikipedia-500K dataset for this experiment. We first quantify how much a label is similar to the rest of the labels in the dataset, we do this by computing the normalized embeddings of each label (from the trained DE model) and searching for the top 10 nearest labels for each label, we use the mean similarity score of the top 10 nearest labels as an indication of how similar this label is compared to rest of the labels. Based on this mean similarity score, we divide our labels into 3 equal bins, where the first bin corresponds to the *most dissimilar* labels and the last bin corresponds to the *most similar* labels. We compute the contribution of labels from each bin to final P@5 numbers for both the pure classifier-based approach (DistilBERTOvA baseline) and pure dual encoder-based approach (DecoupledSoftmax DE). We then plot the relative improvement between the classifier-based approach and the dual-encoder approach, more specifically we plot the relative improvement in P@5 contribution of the dual-encoder method relative to the classifier method. We also perform the same analysis but this time we create label bins for only tail labels (we define a tail label as any label with $\leq 5$ training points

From Figure 8 we can infer that there doesn't seem to be any meaningful pattern when we look at the results of the analysis on all labels but when we look at the analysis just on tail labels there appears to be a trend that the relative improvement is more on similar labels and less on dissimilar labels. This demonstrates that for labels with less training data, dual-encoders can perform relatively better on similar but distinct labels compared to a pure classifier-based approach which treats each label as an atomic entity.

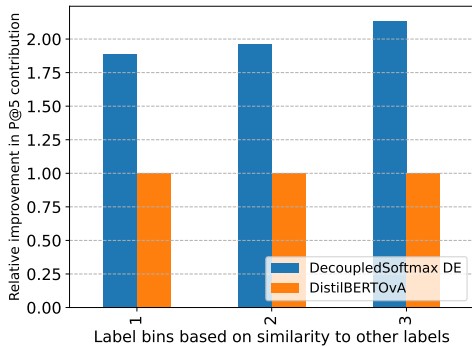 

Figure 8: Plot to study if dual encoder training performs better on similar but distinct labels on LF-Wikipedia-500K dataset. We first divide labels into 3 equal bins, where the first bin corresponds to the *most dissimilar* labels and the last bin corresponds to the *most similar* labels (based on the mean similarity score of the top 10 nearest embedded labels). We then compute the contribution of labels from each bin to final P@5 numbers for both pure classifier-based approach (DistilBERTOvA baseline) and pure dual encoder-based approach (DecoupledSoftmax DE). In the **right** subplot, we plot the relative improvement between the classifier based approach and the dual encoder approach when considering all labels, more specifically we plot the relative improvement in P@5 contribution of the dual encoder method relative to the classifier method. In the **left** perform the same analysis but this time we create label bins for only tail labels (we define a tail label as any label with $\leq 5$ training points.

## C.8 COMPARISON WITH XR-TRANSFORMER ON EURLEX-4K

As highlighted in (Gupta et al., 2022), methods like XR-Transformer use high-capacity sparse classifiers learned on the concatenated sparse tf-idf features and dense embedding obtained from the BERT encoder for ranking their top predictions. In table 15 we compare XR-Transformer's performance without ensembling and without using tf-idf features. We also, report results by adding the sparse ranker layer (implemented

Table 15: Comparison with XR-Transformer on EURLex-4K dataset

| Loss | P@1 | P@3 | P@5 |
|---|---|---|---|
| XR-Transformer ×3 ensemble | 88.41 | 75.97 | 63.18 |
| XR-Transformer | 87.19 | 73.99 | 61.60 |
| XR-Transformer w/o tf-idf ranker | 83.93 | 69.48 | 56.62 |
| DecoupledSoftmax DE | 86.78 | 73.40 | 60.19 |
| DecoupledSoftmax DE with tf-idf ranker | 86.55 | 75.21 | 62.26 |

similarly to the approach described in Section 3.5 of (Gupta et al., 2022)) on top of DecoupledSoftmax DE top 100 predictions. As we can see that XR-Transformer's better performance on EURLex-4K dataset can be greatly attributed to the use of ensemble of 3 learners and high-capacity sparse classifiers.

## C.9 APPLICATIONS OF PROPOSED LOSS FUNCTIONS FOR RAG

Inspired by the open QA benchmarks (NQ/MSMARCO) which can be treated as few-shot variants of RAG task (given a question, retrieve passage using a DE model. then given (question, retrieved passage) a reader model generates answer), we construct a naturally many-shot and multi-label RAG scenario from the existing XMC Wikipedia dataset LF-Wikipedia-500K. More specifically, we create the task of generating Wikipedia content from a Wikipedia page title given some tags associated with the page. The tags are essentially the evidence documents that we retrieve and we use the Wikipedia page title as the query and ask a fine-tuned LLM (Llama2-7B) model to generate the corresponding Wikipedia article. Following are some sample query prompts:

- "Given the associated wikipedia tags [(living people), (american film actresses), (actresses from chicago illinois)], Write a wikipedia like article on [Aimee_Garcia]: '"

- "Given the associated wikipedia tags [(1928 births), (auschwitz concentration camp survivors), (american soccer league 1933 83 coaches)], Write a Wikipedia like article on [Willie_Ehrlich]: "

- "Given the associated wikipedia tags [(given names), (germanic given names), (norwegian feminine given names)], Write a wikipedia like article on [Solveig]:"

We fine-tune the LLM model on 10K training samples and evaluate the LLM augmented with different retrieval approaches on separate 1K test samples. We compare the generated content with the gold content using ROUGE metrics.

| Retrieval | ROUGE-1 | ROUGE-3 | ROUGE-L |
|---|---|---|---|
| No retrieval | 0.1622 | 0.0468 | 0.1236 |
| InfoNCE DE | 0.1806 | 0.0532 | 0.1371 |
| DecoupledSoftmax DE | 0.1838 | 0.0540 | 0.1387 |

Table 16: Performance comparison of different retrieval methods on proposed RAG scenario using ROUGE scores.

Although this is a very preliminary experiment that needs to be thoroughly tested before making any concrete conclusions, it does provide promise for our proposed changes in RAG settings and opens up new applications for XMC approaches.

