# EFFICACY OF DUAL-ENCODERS FOR EXTREME MULTI-LABEL CLASSIFICATION

## ABSTRACT

Dual-encoder models have demonstrated significant success in dense retrieval tasks for open-domain question answering that mostly involves zero-shot and few-shot scenarios. However, their performance in many-shot retrieval problems where training data is abundant, such as extreme multi-label classification (XMC), remains under-explored. Existing empirical evidence suggests that, for such problems, the dual-encoder method's accuracies lag behind the performance of state-of-the-art (SOTA) extreme classification methods that grow the number of learnable parameters linearly with the number of classes. As a result, some recent extreme classification techniques like (Dahiya et al., 2023a;b) use a combination of dual-encoders and a learnable classification head for each class to excel on these tasks. In this paper, we investigate the potential of "pure" DE models in XMC tasks. Our findings reveal that when trained correctly standard dual-encoders can match or outperform SOTA extreme classification methods by up to 2% at Precision@1 even on the largest XMC datasets while being $20\times$ smaller in terms of the number of trainable parameters. We further propose a differentiable top-k error-based loss function, which can be used to specifically optimize for Recall@k metrics. We include our `PyTorch` implementation along with other resources for reproducing the results in the supplementary material.

## 1 INTRODUCTION

Dual-encoder (DE) models have been highly successful in dense retrieval tasks for open-domain question answering (openQA) systems (Lee et al., 2019; Karpukhin et al., 2020; Qu et al., 2021), where they efficiently retrieve relevant passages or documents from a large corpus given a user's query. These models excel at leveraging their learned representations to identify and rank pertinent documents, by mapping both queries and documents into a shared embedding space, enabling efficient and effective retrieval using *fast similarity search methods* (Guo et al., 2020; Johnson et al., 2021). So far, DE models have been primarily explored in the zero-shot and the few-shot scenarios, where the models are required to *generalize* and retrieve relevant documents even though they might not have appeared in the training set.

An important retrieval scenario that frequently arises in real-world applications such as search engines (Mitra & Craswell, 2018) and recommendation systems (Covington et al., 2016; Zhang et al., 2019) is where we want to perform retrieval for documents/items from a large collection based on a significant number of seen examples per document/item, i.e., a *many-shot scenario*. Such tasks are often formulated as extreme multi-label classification (XMC) task, where each document/item to be retrieved is considered as a separate label (Bhatia et al., 2016). Typically, XMC algorithms need to both "memorize" and "generalize". That is, for each label they need to memorize the type of queries that are relevant; e.g., for a product to product recommendation scenario, the algorithm should memorize which products can lead to click on a particular product using the provided product-product co-click data. At the same time, the algorithm should generalize on unseen queries.

It is a prevailing belief in the XMC community that due to the semantic gap and knowledge-intensive nature of XMC benchmarks, DE by themselves are not sufficient to attain good performance (Dahiya et al., 2023b). As a measure to overcome the semantic gap and enable memorization, SOTA XMC methods augment DE with either per-label classifiers (Dahiya et al., 2023a; Mittal et al., 2021a) or extra auxiliary parameters (Dahiya et al., 2023b). We explore this belief by performing a simple

experiment on a synthetic dataset where a random query text is associated with a random document text, and the task is to memorize these random correlations during retrieval. As shown in Section C.1, DE models are able to perform this task with perfect accuracy at least on up to 1M scale datasets, disputing the previously held belief in the literature.

In light of the aforementioned synthetic experiment, this work aims to answer the following question: "Are pure DE models sufficient for XMC tasks?" A Pure DE model is desirable because: 1). For XMC methods, unlike DE, the model size and consequently the number of trainable parameters scales *linearly* with the number of labels (see Figure 1). 2). XMC methods require re-training or model updates on encountering new labels. In contrast, DE methods can generalize to new labels based on their features. Interestingly, our work shows that pure DE models can indeed match or even outperform SOTA XMC methods by up to 2% even on the largest public XMC benchmarks while being 20× smaller in model size. The key to the improved performance is the right loss formulation for the underlying task and the use of extensive negatives to give consistent and unbiased loss feedback.

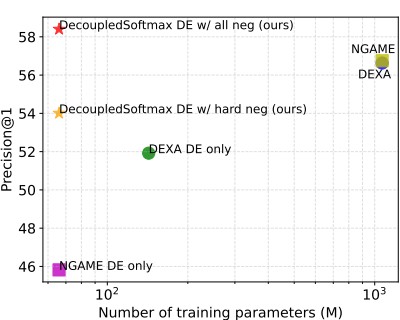

Figure 1: Number of trainable parameters used by different models and their P@1 performance on LF-AmazonTitles-1.3M dataset

Our proposed changes to DE training are based on the observation that XMC problems involve multiple positives per query and a long tail of labels, which at query level manifests as a large number of highly relevant negatives. Standard contrastive learning losses used to train DE models, e.g., InfoNCE loss (Oord et al., 2018), are not well-suited for multi-label learning tasks. Moreover, due to the computational requirements posed by encoding a large number of negative labels per query, standard approaches do not train DE models with sufficient number of negatives, whereas typical extreme classifiers are trained with significantly more ($\mathcal{O}(100)$) negatives per query. We provide a simple fix to InfoNCE loss to allow for multi-label correlation between query and labels by proposing a decoupled loss that decomposes the loss over individual positives. This is required since standard contrastive losses forces all positive labels to have scores in similar range but not all positive labels possess equal ease of prediction from the available data. For example, in EURLex-4K dataset labels "afghanistan" and "international_sanctions" often occur together but "afghanistan" is much easier to infer from the query text than "international_sanctions". In Section 4 we provide a more detailed motivation/justification for our proposed change to the InfoNCE loss. To establish the capabilities of DE methods, we first study our proposed loss using *all* the negatives in the loss term. Naturally, including all negatives is challenging for million-label scale datasets. To this end, we provide a memory-efficient distributed implementation that can use multiple GPUs and a modified gradient cache (Gao et al., 2021) approach to scale DE training to largest public XMC benchmarks even with a modest GPU setup (see Table 2, 9). We further study the performances and implications of approximations of the proposed loss functions using standard hard negative mining approaches. Since many practical settings focus on top-k ranking performance, we further refine our loss design to this end. Based on a soft *differentiable* version of top-k operation, we propose a novel loss function and show it can be integrated easily with standard DE training.