# OpenReview forum: "Dual-Encoders for Extreme Multi-label Classification"
_ICLR.cc/2024/Conference — ICLR 2024 poster_

### Official Review · Reviewer_1qgK · 2023-10-31

**Soundness:** 3 good
**Presentation:** 2 fair
**Contribution:** 3 good
**Rating:** 6
**Confidence:** 3

**Summary:**

Dual-encoder models have achieved substantial success in dense retrieval tasks for open-domain question answering, especially in zero-shot and few-shot scenarios. However, their performance in many-shot retrieval problems, where abundant training data is available, such as extreme multi-label classification (XMC), has received limited attention. Existing evidence indicates that dual-encoder methods tend to underperform compared to state-of-the-art extreme classification methods that scale the number of learnable parameters linearly with the number of classes in such tasks. Some recent extreme classification techniques combine dual-encoders with a learnable classification head for each class to excel in these scenarios. This paper explores the potential of "pure" dual-encoder models in XMC tasks and provides insights for XMC.

**Strengths:**

Important Research Problem: The paper addresses a significant and relevant problem in the field of machine learning - extreme multi-label classification. This is a challenging task with practical applications, and the choice of topic adds value to the existing literature.

Interesting Approach: The idea of using dual-encoders for solving the problem is intriguing and adds a novel dimension to the research. This innovative approach can potentially open up new avenues for tackling similar problems in the future.

Comprehensive Experimental Validation: One of the strengths of the paper is its thorough experimental validation. The fact that the proposed approach has been tested on multiple datasets indicates a comprehensive evaluation of its effectiveness. This enhances the credibility of the findings and their potential applicability to real-world scenarios.

**Weaknesses:**

I do not find obvious flaws. One notable weakness is the lack of clarity in articulating the paper's contributions. It's important for the reader to clearly understand what novel insights or advancements are being offered. The paper should explicitly state the unique contributions and why they matter in the context of extreme multi-label classification. This clarity is crucial for both researchers and practitioners in the field.
To enhance the paper, it would be beneficial to provide a more structured and explicit statement of the research's contribution and significance in introduction section. This will help the readers better grasp the key takeaways from the study. Additionally, the paper could benefit from improved organization and flow to ensure that the reader can easily follow the arguments and findings.

**Questions:**

As in Weaknesses.

---

> ### Author Response · Authors · 2023-11-17
>
> Thank you for your thoughtful review and constructive feedback. We appreciate your positive comments on the importance of the research problem, our approach, and the thorough experimental validation. We have carefully considered your suggestions and propose the following revisions:
>
> **Clarifying Contributions**: Please see our general comment for a more clear discussion on contributions and significance of this work. We acknowledge the need for a clearer articulation of the paper's unique contributions, especially in the introduction. We have revised the introduction to explicitly reflect this (please see the revised pdf)
>
> **Improving Organizational Flow**: We will restructure the paper to enhance the flow and coherence of our arguments and findings. More specifically, taking into account reviewer Smm7’s suggestions, we will introduce a preliminary discussion on inability of BCE loss to train dual encoders and clearly state how we are proposing to overcome the challenges. Please let us know if there are any further suggestions to improve the organization of the paper.
>
> We hope these revisions will improve the manuscript, addressing your concerns and making our contributions clearer to the readers. We are happy to discuss any further questions about the work and would appreciate an appropriate increase in the score if your concerns were adequately addressed.

---

> > ### Comment · Reviewer_1qgK · 2023-11-22
> >
> > Thanks for the detailed responses. I decide to maintain my score (6).

---

### Official Review · Reviewer_jMAd · 2023-11-01

**Soundness:** 3 good
**Presentation:** 2 fair
**Contribution:** 2 fair
**Rating:** 6
**Confidence:** 3

**Summary:**

This paper explores the performance of dual-encoder models in extreme multi-label classification (XMC) tasks. The authors first reveal the shortcomings of traditional dual-encoder training loss, which may over-penalize the correct prediction of "easy-positive" labels. To address this, the Decoupled Multi-label Loss is proposed, aiming to minimize the undesirable correlation between positive labels during training. A memory-efficient training framework is also introduced in this paper.

**Strengths:**

- The authors highlight a neglected problem in the XMC dual-encoder training stage. The proposed Decoupled Loss effectively solves this problem.
- The theoretical part of the paper is well-presented, with Section 3 providing clear symbol definitions.

**Weaknesses:**

- The main motivation for this paper is the imperfect design of current dual-encoder training loss. However, there is a lack of evidence that this has been a general issue for current XMC methods. Most discussions and experiments are designed to compare the Decoupled Loss and regular loss using the authors' own training framework. Some experiments are implemented using a synthetic dataset (Fig. 2) or pre-selected labels (Fig. 3). After reading the entire paper, I believe the proposed loss can solve the mentioned problem, but I am not convinced that this problem is a universal issue in the XMC community.
- The paper lacks novelty. A minor revision of the training loss may not be sufficient for a top conference paper.

**Questions:**

- Please address the questions mentioned in the Weaknesses section.

---

> ### Author Response · Authors · 2023-11-17
>
> We thank the reviewer for the review and feedback. Please find below our response to the concerns:
>
> > The main motivation for this paper is the imperfect design of current dual-encoder training loss. However, there is a lack of evidence that this has been a general issue for current XMC methods...
>
> We disagree that “there is a lack of evidence that this has been a general issue for current XMC methods and most discussions and experiments are designed to compare the Decoupled Loss and regular loss using the authors' own training framework”. We perform our evaluation on 4 XMC benchmarks which span the largest available XMC datasets. The main results of our paper (Table 2 and 3) compare our approach against current SOTA XMC approaches [1,2] and show superior performance. Existing XMC papers repeatedly show that dual encoders don’t perform well by themselves and hence need to be augmented with per-label classifiers or auxiliary parameters to achieve the best performance, see table 8 in [1] and table 2 in [2] vs results in [3] which show upto 10% absolute difference in P@1 between vanilla dual-encoder approach and dual-encoder with classifier approach. Quoting from [2], “this (dual-encoder) approach is expected to suffer if the textual descriptions are not descriptive enough which makes it challenging for an encoder to bring related data points and labels close to each other in the embedding space....To overcome this, XC methods introduce a high-capacity classifier such as 1-vs-all models where each label is endowed with a linear classifier that acts on the embedding of a data point”. In contrast, to the best of our knowledge, our work is the first one to show that vanilla dual-encoders can match or surpass existing SOTA methods when trained appropriately. This is a desirable result because a) dual-encoders can be much more parameter efficient than existing XMC methods that use per-label classifiers; b) this brings promise that dual-encoders can solve both standard retrieval and XMC problems leading to a universally applicable solution and simplifying the landscape of future solutions for these problems. Please see our general comment for a more detailed discussion on contributions and significance of this work. We also acknowledge that although this work shows that dual-encoders are capable of performing well on XMC tasks, classifier based solutions can still be attractive because of their ease of computation (which is just an embedding lookup) as compared to encoders (which requires a relatively expensive encoder forward pass).
>
> > The paper lacks novelty. A minor revision of the training loss may not be sufficient for a top conference paper.
>
> We would like to emphasize that the technical contributions of our paper goes beyond just introducing a minor revision to existing loss. More specifically, our technical contributions include:
> - Identification of Limitations in Traditional Dual-Encoder Training Losses: we analyze existing multi-label and contrastive training losses like One-versus-All Binary Cross-Entropy (OvA-BCE) and InfoNCE, highlighting how they may be inadequate for training dual-encoder models in XMC. Specifically, we note that OvA-BCE does not train effectively and InfoNCE disincentivizes confident predictions.
> - Decoupled Softmax and SoftTop-k loss: To overcome these limitations, we proposes a simple modification to the InfoNCE/Softmax loss as the Decoupled Softmax Loss and extend the loss design to soft-topk loss which is tailored to optimize prediction accuracy within a specific prediction budget size ('k')
> To the best of our knowledge, these limitations of standard multi-label and contrastive loss functions and the proposed changes have not been discussed in literature before. Moreover, scaling our experiments to largest XMC benchmarks require non-trivial implementation efforts which is often not highlighted but is an essential component for modern machine learning solutions.
>
> ---
>
> [1] NGAME: Negative Mining-aware Mini-batching for Extreme Classification, WSDM 2023
>
> [2] Deep Encoders with Auxiliary Parameters for Extreme Classification, KDD 2023
>
> [3] The Extreme Classification Repository: Multi-label Datasets & Code
>
> ---
>
> We hope our response addressed your concerns and are happy to discuss further questions. Please consider improving your score if you find this work adequate for acceptance after the clarifications and revisions.

---

> > ### Comment · Reviewer_jMAd · 2023-11-21
> >
> > Thanks for authors' detailed explanation. I've raised my score to 6.

---

### Official Review · Reviewer_BerB · 2023-11-03

**Soundness:** 3 good
**Presentation:** 2 fair
**Contribution:** 2 fair
**Rating:** 6
**Confidence:** 2

**Summary:**

This paper explores the use of dual encoder models for extreme multilabel classification (XMC). Dual encoder models, which as the name suggests, use two encoder models, have been generally effective for a variety of other types of tasks involving zero-shot and few-shot learning, but have underperformed the state-of-the-art on XMC tasks. Dual encoder methods are, on the other hand, desirable for extreme classification in principle because they can be much more parameter efficient than extreme classification methods whose parameter counts have a strong dependence on the number of classes.

The authors propose a new loss function, the differentiable top-k, for dual encoder models on XMC tasks that make them competitive with the state-of-the-art methods on these tasks--often outperforming them by a significant margin.

**Strengths:**

- The work is well-motivated and addresses the practical problem of getting dual encoder methods to work in the extreme multilabel setting.
- The proposed contribution is simple, as it is just a loss function paired with either a negative mining approach or a memory-efficient implementation using all negatives.
- The ablation of the different loss variants -- soft top-5 and soft top-100 -- is compelling and shows that the method can effectively optimize precision or recall at 5 or 100, respectively.

**Weaknesses:**

- I might have missed something, but I think it should be made clearer earlier on in the paper that the differentiable top-k operator had been proposed previously [1]. The authors also link to the author of the stackexchange answer, but it would be ideal to cite the specific answer at the link (please correct me if this appears somewhere in the main text, but I couldn't find it). Relatedly, is there other work that uses the formulation by Thomas Ahle? For example, how does this formulation compare to the one linked in the stackexchange post [2]? In my opinion, this brings the novelty of the contribution *when posed as a new loss function* into question.
- Again, I might have missed something, but I think the ablation involving the negative mining approach should include a comparison to SOTA methods, as the negative mining approach may be required for scaling the method up even further.

[1] https://math.stackexchange.com/questions/3280757/differentiable-top-k-function/4506773#4506773

[2] https://arxiv.org/pdf/2002.06504.pdf

**Questions:**

- Can the authors elaborate on how $t_x$ in the soft top-k formulation is computed via binary search? As this is a crucial hyperparameter of the loss function, does this require a logarithmic number of retraining runs? How many runs are required to set this, what are the compute costs involved with setting this hyperparameter, and how strongly does the final performance depend on this choice?
- In Table 5, why is precision at 5 reported while recall at 100 is reported? Why not both precision and recall at 5 and 100? Is this a standard choice?
- Does the loss optimize precision at k or recall at k?

---

> ### Author Response · Authors · 2023-11-17
>
> Thank you for your valuable feedback. We appreciate your insights and have addressed your concerns as follows:
> >  I might have missed something, but I think it should be made clearer earlier on in the paper that the differentiable top-k operator had been proposed previously...
>
> You’re right that the application of the differentiable top-k operator as a loss function for optimizing multi-label predictions at fixed prediction budget-size is the novel contribution. The differentiable top-k operator itself has been proposed in the literature, a common approach is to use the optimal transport-based formulation (which you mentioned) but it is very hard to implement in a distributed training setup which is necessary for our experiments. This is why we instead extend the approach presented in the stackexchange answer by Thomas Ahle because of its simplicity and easily distributable operations with low communication overheads. To the best of our knowledge this formulation is not used in any existing work. We do cite the original stackexchange answer but ICLR citation style seems to only display user profile in the main text (the citation in the reference links to the original answer), we will fix this in the revised version based on Reviewer Smm7’s suggestions. We’d also like to highlight that although soft-topk loss function is one of the technical contributions we make, the major contributions of this work is in highlighting that existing popular loss functions can be inadequate for training dual-encoder models for XMC, to overcome this we revise InfoNCE and extend its loss design to SoftTopK leading to much more parameter efficient SOTA XMC solution. Please see our general comment for a more detailed discussion on contributions and significance of this work.
>
> > ablation involving the negative mining approach should include a comparison to SOTA methods
>
> Due to lack of space we did not repeat the rows from the main results table (Table 2) which reports numbers for SOTA methods in the negative mining ablation table (Table 4). We do agree that perhaps having the existing SOTA results in Table 4 might be easier for a reader to quickly compare the approach with negative mining, we will make this change in the final version. In the table below we report the results with negative mining against SOTA methods, note that DE trained with proposed loss functions can still match or come very close to SOTA methods.
> | Method | Hard neg per query | P@1     | P@5     | R@10    | R@100   |
> |--------|-------------------|---------|---------|---------|---------|
> | **LF-Wikipedia-500K** | | | | | |
> | Ours   | 0                 | 77.71   | 43.32   | 69.24   | 88.12   |
> | Ours   | 1                 | 82.85   | 48.84   | 74.47   | 90.08   |
> | Ours   | 2                 | 83.34   | 49.32   | 74.73   | 89.74   |
> | Ours   | 5                 | 83.86   | 49.57   | 74.60   | 89.12   |
> | Ours   | 10                | 84.77   | 50.31   | 75.52   | 90.29   |
> | NGAME (with classifiers) | - | 84.01 | 49.97 | - | - |
> | DEXA (with classifiers) | - | 84.92 | 50.52 | - | - |
> | **LF-AmazonTitles-1.3M** | | | | | |
> | Ours   | 0                 | 42.15   | 32.97   | 29.28   | 57.48   |
> | Ours   | 1                 | 49.16   | 39.07   | 32.76   | 60.09   |
> | Ours   | 2                 | 50.74   | 40.14   | 33.31   | 60.45   |
> | Ours   | 5                 | 52.04   | 40.74   | 33.48   | 60.13   |
> | Ours   | 10                | 54.01   | 42.08   | 34.19   | 61.04   |
> | NGAME (with classifiers) | - | 56.75 | 44.09 | - | - |
> | DEXA (with classifiers) | - | 56.63 | 43.90 | - | - |
>
> > How $t_x$ in the soft top-k formulation is computed via binary search...
>
> $t_x$ is not a hyperparameter, rather $t_x$ is a variable which is a function of the $x$ vector (i.e. the vector of all the scores) such that $\sum_{i=1}^{N}{s(x_i + t_x) = k}$. Essentially, t_x is a scalar quantity which when added to all the scores $x_i$ makes the sigmoid of new scores ($x_i + t_x$) sum upto $k$. Since sigmoid is a monotonic function, increasing $t_x$ strictly increases the sum of sigmoids and decreasing $t_x$ strictly decreases the sum of sigmoids, hence for a given $x$, binary search can be used to find the $t_x$ which makes the sum of new sigmoids equal to $k$. We hope this clarifies the doubt, we will also revise section 4.3 in the final version to make this more clear.

---

> > ### Author Response · Authors · 2023-11-17
> >
> > > Why is precision at 5 reported while recall at 100 is reported?
> >
> > It is a standard practice in recent XMC papers to report precision for lower K (i.e. K=1-5) and report recall for higher K (i.e. K=10-100)[1,2]. For a particular K, precision and recall numbers are proportional to each other as their numerator is same (number of true positives), the only difference is that they are scaled differently, for precision we’ll divide by K but for recall we’ll divide it by total number of positives (which is a fixed quantity), we believe this is why reporting precision at small K and recall at higher K gives better resolution to compare results at different prediction budget size.
> >
> > > Does the loss optimize precision at k or recall at k?
> >
> > For a fixed prediction budget-size K, empirically improving precision@k improves recall@k as well and vice versa since both of these share the same numerator but are only scaled differently in the denominator. The individual per-query loss terms can be scaled accordingly if we strictly want to optimize for precision@k or recall@k.
> >
> > ---
> >
> > [1] ELIAS: End-to-end Learning to Index and Search in Large Output Spaces, Neurips 2022
> >
> > [2] DECAF: Deep Extreme Classification with Label Features, WSDM 2021
> >
> > ---
> > We hope our response addressed your concerns and are happy to discuss further questions. Please consider improving your score if you find this work adequate for acceptance after the clarifications and revisions.

---

> > > ### Comment · Reviewer_BerB · 2023-11-22
> > > **Thank you for the response!**
> > >
> > > Thank you for your response, which has addressed my questions and concerns -- I will raise my score to a 6 and recommend acceptance.

---

### Official Review · Reviewer_Smm7 · 2023-11-06

**Soundness:** 3 good
**Presentation:** 2 fair
**Contribution:** 3 good
**Rating:** 8
**Confidence:** 3

**Summary:**

The manuscript considers the use of Dual Encoders (DE) models for Extreme Multi-Label Classification (XMC). In previous work using softmax loss functions, the presence of other positive classes decreases the negative loss that can be gained from a training example. This work considers decoupled softmax, in which the negative loss gained for each positive class is independent of other positive classes. The motivation for this new loss function is that some classes may "obviously" apply, which should lead to confident predictions for those "obvious" cases. In experiment with synthetic dataset where one of the samples is very obviously marked, both softmax and decoupled softmax methods latch on this obvious annotations early on, but standard softmax approaches then entice the model to give the same score to all positive samples, obvious or not, thus decreasing the model's confidence in the obvious case. In non-synthetic settings, decoupled softmax reduces the variance in gradient feedback.

Directly training with the full decoupled softmax loss is challenging when there are many labels: appendix presents how to do this training with the full loss in a memory-efficient way. Still, for higher number of labels (e.g., documents retrieval), there is a need to approximate this full loss, and some preliminary results are given in this direction. A natural extension is presented for SoftTop-k.

The Appendices were not considered as part of this review.

**Strengths:**

To the best of my knowledge, the work is original and significant. Although the distinction between using a single multi-class cross-entropy vs using multiple binary cross-entropies (OvA-BCE) have long been well understood, it isn't immediately obvious that the DE setting would be so different, and that the OvA-BCE loss would fail to train.

In the context of Retrieval-Augmented Generation (RAG), when the retriever is not trained in an end-to-end manner with the answerer, one often conceptualize the retrieval problem as "get that one good document that is know to suffice to answer the question". However, there could be multiple such documents that contain the desired answer: this retrieval task can be understood as a particularly extreme case of XMC. The take home message that I personally get from this manuscript is that I should exercise caution along those lines if I ever attempt to train DE retrieval models with contrastive augmentations. (continued in weaknesses)...

**Weaknesses:**

... (continued from strengths) However, I wouldn't have come to this realization from the manuscript's abstract nor introduction, and that nugget would have been lost on me had I encountered the manuscript outside of a reviewing context. I understand that this work mainly targets the XMC literature (of which - full disclosure - I personally don't know much), but I still believe that the authors should dedicate some of their high-level discussions (i.e., abstract and/or introduction), to the significance of their work for DE-based RAG.

More generally, the manuscript's main weakness is with organizing the content to give a clear narrative. For example, Equation (2) introduces OvA-BCE, then eschew saying what's wrong with it until Section 5.5. As a reader, I had to go back-and-forth in the document to just understand what the authors are set in doing, what goes wrong with the default approach, and what's the authors' solution. More details in Question 1 below.

**Questions:**

### 1
(a) What are the assumptions, goals and constraints specifying what this project is about; (b) what would be the default/status quo approach and what's wrong with it; and (c) what is the essence of the solution you propose?

I think that I managed to figure out the answer to those questions, but these things should be clearly identifiable from the introduction (or even abstract). Here's my own crude attempt at it: please complement it, clarify any disagreements, and propose edits to the manuscript.

- 1(a) There are a large number of classes, and many of them can apply to the same sample. The models being considered are DE where the representations from each encoder will be converted to a score using an inner product.
- 1(b) OvA-BCE won't train; InfoNCE disincentivizes confident predictions.
- 1(c) Decoupled softmax both trains and allows for confident predictions.


### 2
The bulk of the manuscript presumes the optimization of the the "full" loss function over all positive and negative classes. Is this standard practice in the XMC community? Some experiments sampling hard negatives are presented in Section 5.4, but there is no real discussions besides "more is better". Do you have any insights to add? Could those be added to the manuscript?


*The remaining points are more minor*

### 3
Figure 1: why express the x axis in millions, instead of replacing the $10^2$ and $10^3$ ticks by $10^8$ and $10^9$?

### 4
Please avoid notations such as "O(100)", "O(million)" and "O(billion)".

### 5
Figure 2: "Precision@1" here has a special meaning. There are 5 samples that are marked as positives, but one of them is "more positive", and "Precision@1" here means "the more positive sample must be in first position". This special setting is relatively clear in the text, but looking quickly at Figure 2 and its caption can be misleading. Please consider inventing a different term/notation.

### 6
There are multiple missing punctuations after mathematical expressions. For example, a period should be added at the end of the first paragraph of Section 3, and another one should be added after the equation at the bottom of page 4. As a side note, I would personally add a lot of commas to the English text, but I understand that this may be more a matter of style.

### 7
The text appears to use \cite or \citep everywhere, but many should actually be \citet. For example, in the second paragraph of Section 4.2, "similar to (Xiong et al., 2021)" should become "similar to Xiong et al. (2021)", and "in (Lindgren et al., 2021)" should be "in Lindgren et al. (2021)".

### 8
The style of the stackexchange citation in the paragraph following Equation (6) should be revised. Personally, I would have expected something like "... on a proposal by Ahle (2022)."

### 9
Some variables are overloaded; for example $d$ is used both for documents and dimensions. I thought that there were more cases, but I can't find them anymore, so I may be confused with another paper. Please check.

---

> ### Author Response · Authors · 2023-11-17
>
> Thank you for your thorough and insightful review. We appreciate your recognition of the originality and significance of our work in applying Dual Encoders in Extreme Multi-Label Classification and its implications for Retrieval-Augmented Generation. Please find below our response to the concerns and suggestions raised:
>
> **Implications for RAG**: Retrieval augmented generation (RAG) is an important scenario which we didn’t fully consider at the point of writing this paper, many thanks for bringing it out. In scenarios where multiple documents could contain the desired answer, the retrieval task in RAG aligns closely with extreme multi-label classification (XMC). Our work's exploration of dual-encoder methods in XMC directly informs the appropriate choice for loss formulation in such scenarios. To this end, we perform a preliminary experiment described in “RAG experiment” below, to see the promise of this work in the context of RAG.
>
> **Significance and Contributions**: Please see our general comment for a more clear description of contributions and significance of this work. We acknowledge your concern regarding the lack of clarity in contributions and significance of this work in the introduction of the manuscript. To address this, we have proposed a revision of the introduction (please see the revised pdf). We plan to further refine the introduction in the final version to ensure that the broader implications of our research are clear to a wider audience. Below we slightly complement your answer to the question 1
> - 1(a) There are a large number of classes, and many of them can apply to the same sample. The models being considered are DE where the representations from each encoder will be converted to a score using an inner product.
> - 1(b) DE models empirically don’t perform well on such task but are desirable due to parameter efficiency and generalization to unseen labels; Standard loss function are inadequate - OvA-BCE won't train; InfoNCE disincentivizes confident predictions.
> - 1(c)Decoupled softmax both trains and allows for confident predictions leading to SOTA DE models for XMC tasks
>
> **Narrative Clarity**: We acknowledge that the manuscript would benefit from a clearer narrative structure. We will reorganize the content in the final version to more effectively present our objectives, the limitations of existing approaches, and the essence of our proposed solution. More specifically, in order to address the concern related to the lack of discussion on OvA BCE in section 4, we will introduce a preliminary discussion on the inability of BCE loss to train dual encoders and clearly state how we are proposing to overcome the challenges. We will also take into account all the formatting suggestions to improve the final version.
>
> **Discussion on negatives**: optimization of the "full" loss function is not a standard practice in the XMC community but is used sometimes to establish the upper bound performance. In XMC benchmarks, we notice that an extensive use of negatives is required to train properly and is typical for classifier based methods to use ~100 negative labels per training query. In dual encoder training since the most expensive step is the forward and backward pass through the encoder, considering ~100 negative labels per training query for a batch size of 8192 would be computationally almost equivalent to computing the full loss for a dataset with upto 1M labels. Apart from showing that there is a significant dependance of the performance with the number of negatives, in section C.3 of the appendix we note that the quality of negatives can also play a modest role in determining the accuracy (specially P@1) performance of the model. In section C.2.3 we also confirm that increasing negatives for InfoNCE/Softmax loss function on XMC benchmarks doesn’t benefit the performance that much when compared to DecoupledSoftmax loss, this is because as we increase total number of negatives to consider in the loss computation we also increase the undesired competition among the positives in a multi-label setting.
>
> ## RAG experiment
> Inspired by the open QA benchmarks (NQ/MSMARCO) which can be treated as few-shot variants of RAG task (given a question, retrieve passage using a DE model. then given (question, retrieved passage) a reader model generates answer), we construct a naturally many-shot and multi-label RAG scenario from the existing XMC wikipedia dataset LF-Wikipedia-500K. More specifically, we create the task of generating wikipedia content from a wikipedia page title given some tags associated with the page. The tags are essentially the evidence documents that we retrieve and we use the wikipedia page title as the query and ask a fine-tuned LLM (Llama2-7B) model to generate the corresponding wikipedia article. contd. in further comment...

---

> > ### Author Response · Authors · 2023-11-17
> >
> > Sample query prompts:
> > - “Given the associated wikipedia tags [(living people), (american film actresses), (actresses from chicago illinois)], Write a wikipedia like article on [Aimee_Garcia]:  '”
> > - “Given the associated wikipedia tags [(1928 births), (auschwitz concentration camp survivors), (american soccer league 1933 83 coaches)], Write a wikipedia like article on [Willie_Ehrlich]:  ”
> > - “Given the associated wikipedia tags [(given names), (germanic given names), (norwegian feminine given names)], Write a wikipedia like article on [Solveig]:”
> >
> > We fine-tune the LLM model on 10K training samples and evaluate the LLM augmented with different retrieval approaches on separate 1K test samples. We compare the generated content with the gold content using ROUGE metrics.
> >
> > |Retrieval | ROUGE-1 | ROUGE-3 | ROUGE-L |
> > |----|----|----|----|
> > | No retrieval | 0.1622 | 0.0468 | 0.1236 |
> > | InfoNCE DE | 0.1806 | 0.0532 | 0.1371 |
> > | DecoupledSoftmax DE | 0.1838 | 0.0540 | 0.1387 |
> >
> > Although this is a very preliminary experiment that needs to be thoroughly tested before making any concrete conclusions, it does provide promise for our proposed changes in RAG settings and opens up new applications for XMC approaches.
> >
> > ---
> >
> > We hope these revisions and additions will address your concerns and significantly enhance the manuscript. Thank you again for your valuable feedback, we are happy to answer any further questions and would appreciate an appropriate increase in the score if the concerns are adequately addressed.

---

> > > ### Comment · Reviewer_Smm7 · 2023-11-22
> > >
> > > The Authors addressed most of my points, they clarified some points on which I was confused, their new introduction is already much better than it was, and they "will introduce a preliminary discussion on the inability of BCE loss to train dual encoders and clearly state how we are proposing to overcome the challenges [and] will also take into account all the formatting suggestions to improve the final version."
> > >
> > > I would raise my score to 7 but it is not among the options, so I raise it to 8 and recommend acceptance.

---

### Author Response · Authors · 2023-11-17
**Contributions and significance**

As both reviewers Smm7 and 1qgK have suggested that there is a lack of clarity in articulating the paper's contributions, we give a clearer description of the contributions and significance of this work here. We have also revised the introduction section to address this.

## What?
This work explores the effectiveness of dual encoders in extreme multi-label classification (XMC), traditionally challenged by the need for per-label classifiers. It demonstrates that dual encoders can match or outperform state-of-the-art methods without relying on per-label classifiers.

## Why/Significance?

- **Parameter Efficient XMC approach**: Existing XMC methods, use per-label classifiers and tend to become increasingly complex and resource-intensive as the number of classes grows. By demonstrating that dual-encoder methods alone can achieve state-of-the-art performance, this work suggests a more parameter efficient approach, especially vital as the number of potential labels in real-world applications continues to grow. Moreover, dual-encoders naturally allow XMC applications to generalize to new unseen labels.

- **Unified Methodological Approach**: The research bridges the gap between XMC and retrieval problems, traditionally treated as similar but distinct problems. By showing the effectiveness of dual-encoders for XMC, it paves the way for developing methods that are universally applicable, simplifying the landscape of machine learning solutions for these problems.

- **Implications for RAG**: In scenarios where multiple documents could contain the desired answer, the retrieval task in retrieval augmented generation (RAG) aligns closely with extreme multi-label classification (XMC). The work's exploration of dual-encoder methods in XMC directly informs the appropriate choice for loss formulation in such scenarios. In response to reviewer Smm7, we perform an additional experiment to investigate preliminary promise of our work for RAG tasks.


## Technical contributions

- **Identification of Limitations in Traditional Dual-Encoder Training Losses**: we analyze existing multi-label and contrastive training losses like One-versus-All Binary Cross-Entropy (OvA-BCE) and InfoNCE, highlighting how they may be inadequate for training dual-encoder models in XMC. Specifically, we note that OvA-BCE does not train effectively and InfoNCE disincentivizes confident predictions.

- **Decoupled Softmax and SoftTopk Loss**: To overcome these limitations, we propose a simple modification to the InfoNCE loss as the Decoupled Softmax Loss. This loss is designed to minimize the undesirable correlation between positive labels during training. We further extend our loss design to Soft-Topk loss which is tailored to optimize prediction accuracy within a specific prediction budget size ('k'). This can be particularly relevant to XMC applications where the goal is to make a limited number of highly accurate predictions from a vast set of possible labels.

To the best of our knowledge, these limitations of standard multi-label and contrastive loss functions and the proposed changes have not been discussed in the literature before. Moreover, scaling our experiments to the largest XMC benchmarks requires non-trivial implementation efforts which is often not highlighted but is an essential component for modern machine learning solutions.

---

### Author Response · Authors · 2023-11-21

We sincerely thank all reviewers for their insightful feedback. We have tried our best to answer the concerns raised and are happy to clarify/discuss any further questions. As the discussion period nears its end, please let us if you have any additional questions or if our responses have resolved your concerns.

---

### Meta-Review · Area_Chair_azGm · 2023-12-04

**Metareview:**

This paper studies the use of dual-encoder models to tackle extreme multi-label classification settings. Specifically, an empirical assessment is carried out with results suggesting that, contrary to what was observed in previous work, dual-encoder architectures can match or outperform existing methods in a number of benchmarks upon modifications in the training objective. Reviewers were consistent in recognizing the relevance of the work and highlighting it's simplicity and usefulness and, while the original version of the manuscript focused on multi-label classification under very large label sets, later results obtained during the discussion further showed promise on more standard retrieval settings leveraging the proposals from the paper. We thus recommend acceptance.

**Justification For Why Not Higher Score:**

The scope of the paper is not broad enough and its impact is limited to a very specific problem. I would argue for a higher score had the experiments on RAG for instance been a bit more extensive and conclusive. I do think however that the paper would be a good addition to the program.

**Justification For Why Not Lower Score:**

N/A.

---

### Decision · Program_Chairs · 2024-01-16

Accept (poster)